



# Estimation of the terms acting on local surface one-hour temperature variations in Paris region: the specific contribution of clouds

Oscar Rojas[1], Marjolaine Chiriaco[1], Sophie Bastin[1], Justine Ringard[1]

[1] LATMOS/IPSL, UVSQ, Université Paris-Saclay, Sorbonne Université, CNRS, 78280, Guyancourt, France

*Correspondence to*: Oscar Rojas (oscar.rojas@latmos.ipsl.fr)

**Abstract.** Local temperature variations at the surface are mainly dominated by small-scale processes coupled through the surface energy budget terms, which depend mostly on radiation availability and thus cloud processes. A method to determine each of these terms based almost exclusively on observations is presented in this paper, with the main objective to estimate

their importance in hourly surface temperature variations at the SIRTA observatory, near Paris. Almost all terms are estimated from the multi-year dataset SIRTA-ReOBS, following a few parametrizations. The four main terms acting on temperature variations are radiative forcing (separated into clear-sky and cloud radiation), atmospheric heat exchange, ground heat exchange, and advection. Compared to direct measurements of hourly temperature variations, it is shown that the sum of the four terms gives a good estimate of the hourly temperature variations, allowing a better assessment of the contribution of each

term to the variation, with an accurate diurnal and annual cycles representation, especially for the radiative terms. A random forest analysis shows that whatever the season, clouds are the main modulator of the clear sky radiation for 1-hour temperature variations during the day, and mainly drive these 1-hour temperature variations during the night. Then, the specific role of clouds is analyzed exclusively in cloudy conditions considering the behavior of some classical meteorological variables along with lidar profiles. Cloud radiative effect in shortwave and longwave and lidar profiles show a consistent seasonality during

the daytime, with a dominance of mid- and high-level clouds detected at the SIRTA observatory, which also affects surface temperatures and upward sensible heat flux. During the nighttime, despite cloudy conditions and having a strong cloud longwave radiative effect, temperatures are the lowest and are therefore mostly controlled by larger-scale processes at this time.

## 1 Introduction

Regional climate variability is to the first order driven by large-scale atmospheric conditions. In Western Europe, the North Atlantic Oscillation (NAO; Trigo et al., 2002), which is associated with the locations and intensities of the centers of the Iceland low and the Açores high, controls the air mass advection over western Europe and explains a large part of variability. Temperature and pressure conditions are then modulated by the complex terrain (Mediterranean sea, topography, surface heterogeneities): extreme events and temperature anomalies are generally not exclusively explained by the presence of these



large-scale air mass circulations (Vautard and Yiou, 2009). It is, therefore, necessary to consider small-scale processes such as surface-atmosphere interactions and cloud feedbacks to better explain surface temperature variations (e.g. Chiriaco et al., 2014).

Temperature variations at the surface are related to the Surface Energy Balance (SEB) following surface-atmosphere interactions and solar radiation (Wang and Dickinson, 2013) that are separated into different components: latent and sensible

heat fluxes ($F_{lat}$ and $F_{sens}$, respectively), ground heat flux, air advection, and atmosphere radiation. Studies have been made to parametrize these terms when direct measurements are not available (Miller et al. 2017; Arnold et al. 1996) but their exact contribution is uncertain. Depending on the time-scale considered, the importance of each SEB term on temperature variations will change (Bastin et al. 2018; Ionita et al. 2015).

The first objective of the current paper is to quantify the specific local contribution of the primary SEB terms acting on short-

term (i.e. hourly) temperature variations in a Western European location and to determine their importance and the conditions when a certain term predominates over the others, based on a simple linear model. The current study is inspired by Bennartz et al. (2013) who implemented a temperature variation model to study the influence of low-level liquid clouds over the Arctic on the ice surface melt period in July 2012, by estimating all of these SEB terms for a case study. Here, the same approach is used for a different location (mid-latitude) and on a long-time period to get robust statistics. To do so, the study is based on

direct measurements from the SIRTA-ReOBS dataset (Chiriaco et al., 2018), which includes many variables collected since 2002 at SIRTA (Site Instrumental de Recherche en Télédetection Active; Haeffelin et al. 2005), an observatory located in a semi-urban area in the Southwest of Paris, France. This dataset is well suited to the current study objective because: (i) it allows the use of a multi-variable synergistic compilation to study and compare different characteristics of the atmosphere or the surface (e.g. Bastin et al. 2018; Dione et al. 2017; Cheruy et al. 2013, Chiriaco et al., 2014), (ii) it is located in Western

Europe and allows access to the hourly time scale, i.e. the scales of the local processes of the current study. The SIRTA-ReOBS dataset, along with some variables retrieved from ERA5 Reanalysis, enables the development of a model to estimate all the terms involved in the surface temperature variations using predominantly real observations. The weight of each term of the SEB on hourly temperature variations is analyzed using a powerful random forest analysis whose maximal advantage is its capacity to handle up to thousands of input variables and identify the most significant ones.

The main result of this first objective is the fact that clouds are the main modulator on hourly temperature variations for most of the hours of the day and all the seasons. Hence the second objective of the current study is to understand the specific role of clouds and their associated characteristics on hourly temperature variations. Indeed, the influence of clouds on the temperature at the surface can vary depending on their physical properties and the altitude where they are formed (Hartmann et al., 1992; Chen et al., 2000). In general, low-level clouds tend to cool the surface, whereas high-level clouds, such as cirrus

clouds, tend to warm it by absorbing a significant amount of Earth's outgoing radiation. The average contribution of clouds is to decrease the surface temperature, combined with the damping effects of soil moisture, from a global point of view, by reflecting the solar radiation to space. But their damping effects vary depending on the season and the state of the atmosphere. For instance, the reduction in surface temperature is the highest in fall in northern mid-latitudes for specific cases when





precipitation is not significant (Dai et al. 1999). The local contribution of clouds to temperature variations is thus an important

topic to assess how this intake affects local climate variability and extreme local events. Several studies were made primarily focusing on the large scale effects of clouds on the radiative energy balance on a global scale, either at the top of the atmosphere (Arkin and Meisner, 1987; Raschke et al., 2005; Dewitte and Clerbaux, 2017; Willson et al., 1981; Allan et al., 2014; Cherviakov, 2016) or at the surface (Wild et al., 2015; Hakuba et al., 2013; Wild, 2017) or both at the same time (Hartmann, 1993; Kato et al., 2012; Li and Leighton, 1993), but a lack of studies investigating their impact on a smaller scale is noted due

to limited reliable ground-based measurement availability. In this study, to understand how clouds influence the one-hour temperature variations, cases with a predominance of cloud effect on temperature variations are analyzed specifically with lidar profiles to understand the role of the vertical distribution of clouds and their radiative effect on the state of the atmosphere. To answer these two objectives, the current paper is organized as follows: the dataset is presented in Section 2. Section 3 (and Appendix A) consists of describing how the different terms acting on hourly temperature variations are estimated and

evaluating how well the model fits the hourly observations based on statistics. Section 4 presents an assessment to determine which term dominates at different times by performing a random forest analysis firstly for day and night cases, and then separated in a diurnal cycle perspective, along with a mean monthly-hourly and annual cycles analysis of the contribution of each term. Section 5 focuses on a discussion to study the specific role of clouds and the atmospheric conditions under which they develop by assessing only the cloudy cases, which gives an overview of the type of clouds and surface conditions damping

or enhancing surface temperature for both day and nighttime. Section 6 draws conclusions and provides perspectives opened by this work.

## 2 Data

### 2.1 Data used for the temperature variations estimation model

This study is mainly based on the SIRTA-ReOBS dataset analysis. The SIRTA observatory (Haeffelin et al. 2005) is located in a semi-urban area 20-km Southwest of Paris (48.71° N, 2.2° E) and has collected long-term meteorological variables since 2002. The ReOBS project aims to synthesize all observations available at a single observatory at an hourly time scale with an exhaustive data-quality control, calibration, and rigorous treatment, into a single NetCDF file. The SIRTA-ReOBS dataset (Chiriaco et al. 2018, https://doi.org/10.5194/essd-10-919-2018) contains more than 60 variables.

All the necessary hourly variables requested by the current study are available for the period going from January 2009 to February 2014 and are listed in Table 1, allowing the performance of a multi-year analysis. Some variables that were retrieved from measurements at the SIRTA observatory sometimes present gaps due to instrumental issues (Chiriaco et al. 2018; Pal and Haeffelin 2015). In particular $F_{lat}$ and $F_{sens}$ variables are limited (with only 5% and 73% availability, respectively): Sect. 3 and Appendix A show how this issue is handled.





Figure 1 shows the complete dataset availability in the SIRTA-ReOBS dataset for the five-year study period. The availability of data is quasi-homogeneous around 60% for all hours (Figure 1a).  Gaps are most of the time due to the absence of the Mixing Layer Depth (MLD) variable for some complete days, extended sometimes for more than two months (not shown), restricting 71% of MLD data availability. Other gaps are caused by the absence of radiative variables (9% of missing data, see Table 1). Summer is the season with the best data coverage (70%) and winter with the least data coverage (56% - Figure 1b),

whose absences are due precisely to gaps in the MLD variable.

To complete this dataset, ERA5 hourly Reanalysis (spatial resolution of 31 km and 137 levels up to 1 hPa; ERA5: Fifth generation of ECMWF atmospheric reanalyses of the global climate, 2020) is used to estimate the horizontal wind at 10 m (useful to estimate the advection – see Section 3) and the temperature at the mixing layer depth ($T_{MLD}$ - see Section 3.1 and Appendix).

**2.2 Variables used for the cloud contribution analysis**

In order to get vertical information about clouds (see Section 5), SIRTA-ReOBS lidar profiles are also used, based on hourly lidar Scattering Ratio altitude-intensity histograms, calculated as follows:

$$SR(z) = \frac{ATB(z)}{ATB(z)_{mol}}$$   (1)

Where ATB(z) is defined as the total attenuated backscatter lidar signal and ATB(z)$_{mol}$ is the signal in clear-sky conditions.

These altitude-intensity histograms are used to estimate the mean cloud fraction percentage at a given altitude level *z*. The intensity axis which contains SR(z) thresholds as well as the three vertical atmospheric layers (i.e. low, middle and high-level layers) for cloud detection and characterization are defined in Chepfer et al. (2010). The value of SR(z) = -999 corresponds to non-normalized profiles, the value -777 represents the profiles that cannot be normalized due to the presence of a very low opaque cloud and the value of -666 is set as invalid data. Then, bins located in the range 0.01<SR(z)<1 are for clear-sky

conditions, 1.2<SR(z)<5 is defined as unclassified data, and for cloud detection, a threshold of SR(z)>5 is set. Details are given in Chiriaco et al. (2018). Even though the instrument does not operate uninterrupted (it does not operate when it is raining, when it's nighttime, and on the weekends), it is very powerful to get information on the vertical structure of the atmosphere and so understand the category of clouds acting on the temperature variations.

**3 Estimation of the terms acting on surface temperature variations**

The exchange of energy between the surface and the overlying atmosphere involves four terms: radiation, heat exchange with ground, heat exchange with the free atmosphere, and advection. In this section, a model which estimates these four terms is presented, based on several meteorological observations retrieved from the SIRTA-ReOBS dataset. Each term involved in this model is described (further descriptions are presented in Appendix), and then a statistical evaluation is performed to assess how well the model follows the real observations of temperature variations. Finally, the mean monthly-hourly and annual





cycles of the averaged contribution of each term and the same for temperature 1-hour variations ($\frac{\partial T_{2m}}{\partial t}_{obs}$) are presented (split

into day and night).

### 3.1 Model description

The temperature variation at the surface is estimated from the sum of four terms:

$$\frac{\partial T_{2m}}{\partial t} = R + HG + HA + Adv \tag{2}$$

Where R is the net radiative flux at the surface, HG is the ground heat exchange, HA is the atmospheric heat exchange, and
Adv is the air advection. These four terms control the changes in temperature over the surface, but depending on the temporal
scale, one will dominate over the others (Ionita et al., 2012). Following the Appendix and partly based on Bennartz et al.
(2013), these four terms are expressed as:

$$R = \frac{\alpha+1}{\rho c_p MLD} \Delta F_{NET} \tag{3}$$

$$HG = \frac{T_s - T_{2m}}{\tau_s} \tag{4}$$

$$HA = \frac{T_{MLD} - T_{2m}}{\tau_a} \tag{5}$$

$$Adv = \left( u_{10} \frac{\partial T_{2m}}{\partial x} + v_{10} \frac{\partial T_{2m}}{\partial y} \right) \tag{6}$$

Then:

$$\frac{\partial T_{2m}}{\partial t} = \frac{\alpha+1}{\rho c_p MLD} \Delta F_{NET} + \frac{T_s - T_{2m}}{\tau_s} + \frac{T_{MLD} - T_{2m}}{\tau_a} - \left( u_{10} \frac{\partial T_{2m}}{\partial x} + v_{10} \frac{\partial T_{2m}}{\partial y} \right) \tag{7}$$

Where $T_{2m}$ is the surface temperature, $t$ is the time (in hours), $\alpha$ is a coefficient characterizing the form of the temperature
vertical profile in the boundary layer, $\rho$ is the average air density of the boundary layer, $c_p$ is the specific heat of air, MLD is
the mixing layer depth, $\Delta F_{NET}$ is the net radiative flux at the surface, $T_s$ is the temperature in the ground at 20 cm depth, $T_{MLD}$
is the temperature at the top of the boundary layer (or mixed layer depth), $u_{10}$ and $v_{10}$ are the zonal and meridional wind
components respectively at 10 m above the ground, and $\tau_s$ and $\tau_a$ are defined as relaxation timescales for heat exchange

processes in the ground and the atmosphere, respectively.

If the model is realistic, then the sum of the four terms in the left side of Eq. (2), denoted as $\frac{\partial T_{2m}}{\partial t}_{mod}$, is as close as possible to

the observed temperature variations denoted as $\frac{\partial T_{2m}}{\partial t}_{obs}$.

The net radiative flux at the surface is calculated as the difference between the radiative longwave (LW) and shortwave (SW)
fluxes leaving and arriving at the surface as follows:

$$\Delta F_{NET} = F_{SW}^{\downarrow} - F_{SW}^{\uparrow} + F_{LW}^{\downarrow} - F_{LW}^{\uparrow} \tag{8}$$

Where the upward and downward arrows represent the upwelling and downwelling radiation, respectively. It is possible to add
and subtract the clear-sky (CS) downwelling radiation fluxes to Eq. (8), to obtain the flux radiative components in CS and
cloudy (CL) conditions (i.e. the specific contribution of clouds), as follows:





$$\Delta F_{NET} = F_{SW}^{\downarrow} - F_{SW}^{\uparrow} + F_{LW}^{\downarrow} - F_{LW}^{\uparrow} + F_{SW,CS}^{\downarrow} - F_{SW,CS}^{\downarrow} + F_{LW,CS}^{\downarrow} - F_{LW,CS}^{\downarrow}$$

$$\Delta F_{NET} = \underbrace{F_{SW}^{\downarrow} - F_{SW,CS}^{\downarrow} + F_{LW}^{\downarrow} - F_{LW,CS}^{\downarrow}}_{\Delta F_{NET,CL}} + \underbrace{F_{SW,CS}^{\downarrow} - F_{SW}^{\uparrow} + F_{LW,CS}^{\downarrow} - F_{LW}^{\uparrow}}_{\Delta F_{NET,CS}} \qquad (9)$$

Here, the $F_{LW}^{\uparrow}$ flux is the same for clear-sky and cloudy conditions, because it mostly depends on the surface temperature (i.e.

the Stefan-Boltzmann law, $F^{\uparrow} = \sigma T_{2m}^4$, where $\sigma$ is the Stefan-Boltzmann constant), but note that in an annual global mean,

$F_{LW}^{\uparrow} > F_{LW,CS}^{\uparrow}$ (around 0.5 W m$^{-2}$; Allan 2011) due to the increase in longwave radiation emitted toward the surface by clouds,

where a small proportion of this radiation is reflected by the surface.

Hence Eq. (2) becomes:

$$\frac{\partial T_{2m}}{\partial t} = R_{CL} + R_{CS} + HG + HA + Adv \qquad (10)$$

With $R_{CL} = \frac{\alpha + 1}{\rho c_p MLD} \Delta F_{NET,CL}$ and $R_{CS} = \frac{\alpha + 1}{\rho c_p MLD} \Delta F_{NET,CS}$

This formulation specifically allows estimation of the role of clouds ($R_{CL}$) in temperature variations, compared to the other

terms. Details on how each term is estimated along with the assumptions that have been made are stated in the Appendix.

As shown by Eq. (3) to Eq. (6) and in the Appendix, temperature variations are estimated from variables described in Section

2, i.e. predominantly based on observations retrieved from the SIRTA-ReOBS, but also on ERA5 datasets.

Concerning the advection term, the computation of this term requires the extraction of the temperature at 2 m ($T_{2m}$) and

northward ($v_{10m}$) and eastward ($u_{10m}$) wind components at 10 m, at the SIRTA grid point and the surrounding grid points.

The temperature at the mixing layer depth $T_{MLD}$ is retrieved using both SIRTA-ReOBS and ERA5 datasets. First,

radiosoundings available in SIRTA-ReOBS (twice a day) are used to get the pressure at the MLD level and then $T_{MLD}$ is

retrieved from the vertical temperature profile in ERA5 at the nearest grid box from SIRTA Observatory (48.7° N and 2.2° E)

and at this pressure level but at the closest time. Note that uncertainty remains in $T_{MLD}$ because it is the temperature at a certain

pressure level, which is only available twice a day.

### 3.2 Statistical evaluation of the model

Statistics are considered to assess how close the total model ($\frac{\partial T_{2m}}{\partial t}_{mod}$) is to the observed temperature variations ($\frac{\partial T_{2m}}{\partial t}_{obs}$).

The PDFs of each term in Eq. (2) are shown in Figure 2a. Firstly, the radiative term R is dominating and contributes the most

in the temperature variations at an hourly scale since its distribution presents different significant peaks for negative and

positive values of temperature variations. In clear-sky conditions, the radiative term contributes the most to warm the surface

whereas the radiative effect of clouds has an opposite effect. A negative significant contribution by the HA term is also

observed (green line), meaning that the mixing with an atmosphere of higher levels contributes to cooling the surface

atmosphere. The two other terms (HG and Adv) have a similar but weak impact on the temperature variation model at this





time scale. In addition, the PDF of the $\frac{\partial T_{2m}}{\partial t}_{mod}$ is compared to $\frac{\partial T_{2m}}{\partial t}_{obs}$ in one hour in Fig. 2b: differences occur for cases where

the temperature decreases during the hour, but the difference remains low. The modeled PDF fits very well with the observed
PDF for cases where temperature increases during the hour (a smaller number of cases as shown by the PDFs).

Figure 2c shows the scatter plot of the observations versus the model. The correlation between the two datasets is good (0.79)
and the bias remains small (-0.20 °C h$^{-1}$). However, the model has difficulties in reproducing the extremes of temperature
change within an hour, e.g. -6 °C h$^{-1}$ which corresponds to the 5$^{th}$ of July 2011 at 20:00 UTC, or -8 °C h$^{-1}$ which corresponds

to the 2$^{nd}$ of July 2010 at 15:00 UTC. This high decrease of temperature could be associated with a cold pool event, which is
more detectable in summer thanks to higher surface temperatures than those in winter, and bring along with it storms and
heavy precipitations, as detected for these two cases (not shown). The hourly values in the observations (as well as in ERA5)
do not allow the capture of this cold pool event properly in the model developed, since it is related to rapid cloud formation
(<1 h) that is not well captured by the R$_{CL}$ term. However since this type of event is rare and ephemeral on a local scale (Llasat

and Puigcerver, 1990; Conangla et al., 2018), its presence does not bias significantly the general performance of the model.
These statistics are estimated for each season as presented in Table 2. Correlation is high (0.82) and bias is low in summer.
Nevertheless, the standard deviation remains high for this season, probably due to the higher values of temperature variation
and the contribution of all terms involved for the summer in comparison to other seasons. For the spring and fall seasons, the
correlation coefficient is still high (0.80). However, in winter the correlation coefficient is the lowest (0.67) with a higher bias

(-0.31 °C h$^{-1}$). Removing the extreme values of the observations and the model (i.e. taking the values that are within the 5th
and 95th percentile) gives better statistics (values in brackets in Table 2), confirming that the model encounters difficulties in
reproducing these extreme events.

Figure 2d presents the hourly evolution of the five terms on the right side of Eq. (10), their sum (i.e. $\frac{\partial T_{2m}}{\partial t}_{mod}$ in blue solid line)

and $\frac{\partial T_{2m}}{\partial t}_{obs}$ (pink solid line) for the first days of September 2010. As expected, a diurnal cycle is identified for the radiative

(both cloud and clear-sky terms) and the heat atmosphere exchange terms. For the radiative terms, it is quite expected to have
a positive (negative) contribution during daytime (nighttime) due to the presence (absence) of solar radiation. In addition, there
are some days when R$_{CL}$ (dotted red lines) plays an important modulating role in temperature variations, noticed during the
day on September 6$^{th}$ and 8$^{th}$, when clouds manage to provide a maximum cooling effect of -2.9 °C h$^{-1}$.

Evaporation and thermal conduction at the surface increase as the day goes on and the larger these terms are the more they

will prevent the increase in temperature linked to the sensible and latent turbulent fluxes. But overall the temperature increases
during the day when these fluxes increase. However, the atmospheric heat exchange term (HA) is on average a negative
contribution to temperature variations in one hour when F$_{lat}$ and F$_{sens}$ increase in the afternoons. This negative contribution
could be mostly associated with the mixing of masses of air at the top of MLD, where depending on the state of temperature
inversion, $T_{MLD}$ is likely to be very negative and this cold air could cool the surface. When R$_{CL}$ presents an important

contribution, HA becomes weaker and its diurnal cycle is attenuated due to the absence of solar radiation and hence fewer



surface fluxes are developed. The ground heat exchange and the advection terms play a minor role at this time scale and contribute negligibly to the model without a significant diurnal cycle detected.

Finally, the model follows well the observations, with a better agreement for daytime than for night-time, and with a better correlation for the first case (not shown). Reasons that may explain the bias are 1) the limited availability of $T_{MLD}$ data which

is estimated using only two radiosoundings per day and not hourly continuous values and 2) the hypothesis and assumptions in some variables and atmospheric conditions outlined in the Appendix. Further, the temperature variations at night might be hard to quantify accurately due to the minor contribution of each of the non-radiative terms, for which contributions are close to zero, especially for the HA term, and where $F_{lat}$ and $F_{sens}$ are most of the time very low (Section 3.3).

**3.3 Annual- and Monthly-hourly cycles of the different terms**

The mean contributions of each term are averaged monthly for both day and night (Fig. 3), whereas Figure 4 presents the magnitudes of the monthly-hourly mean values from January 2009 to February 2014 of each term of the model. A residual term is calculated on these figures, which is calculated as the difference between the model (sum of all terms) and the observations (i.e. $\frac{\partial T_{2m}}{\partial t}_{mod} - \frac{\partial T_{2m}}{\partial t}_{obs}$).

According to Figure 3b and 4b for nighttime, the clear sky and cloud radiative forcing terms dominate in magnitude the hourly

temperature variations at the surface and the other terms remain close to zero. Still during the night, $\frac{\partial T_{2m}}{\partial t}_{obs}$ monthly mean is negative throughout the entire year, with higher negative values during warm months than during winter. Indeed, $R_{CL}$ is always positive during the night, and has an annual cycle more pronounced than the other terms, having a stronger effect in the cold months, due to the increase in cloud cover, especially low-level clouds which enhance an increase of the downwards flux of LW radiation and whose effect gets weaker while approaching summer, due to the modification of the radiative effect of clouds

for this season. This decrease of $R_{CL}$ in summer explains the $\frac{\partial T_{2m}}{\partial t}_{obs}$ annual cycle during night.

A diurnal cycle is pronounced for all the terms (Figure 4). $R_{CS}$ contributes the most in magnitude to local temperature variations during daytime (Figure 3a), and the other terms damp its effect by providing a negative contribution. Furthermore, all the terms are mainly driven by the solar radiation intensity during the day. The diurnal and annual cycles of the $R_{CS}$ ($R_{CL}$) term are important, by having a positive (negative) contribution during the day, mostly dominating the temperature variations at this

scale (as stated in Gaevskaya, 1962). The contribution of $R_{CL}$ on $\frac{\partial T_{2m}}{\partial t}_{obs}$ during daytime is more important during May and June, and finds its minimum for December and January, when the solar radiation is weak, thus preventing clouds to strongly reduce it. Further, a maximum negative (positive) contribution to temperature variations of $R_{CL}$ ($R_{CS}$) is found during the late mornings for all the months, cooling (warming) the surface with a maximum mean value of -1.3 °C h$^{-1}$ (3.2 °C h$^{-1}$) in May and June at 10:00 UTC and 11:00 UTC.

In addition, the HA term plays an important role in temperature variations for spring and fall seasons, when there is an increase in hourly solar radiation (Fig. 3a and 4d). These two seasons are characterized by an important contribution of this term in the



late morning and the afternoon, especially during spring, with an hourly maximum averaged contribution of -1.1 °C h⁻¹. The difference between day and night for the HA term is the largest during these months when the MLD can reach high values in the afternoon due to increased turbulence. For winter, its contribution is minimal due to the low development of the MLD.

This result is in contrast with the negative (and very low) contribution of the HG term, which is maximal in summer in the afternoon with an average smaller value of -0.25 °C h⁻¹ when the surface temperature is the highest and thus strengthens its cooling action during daytime. An opposite effect is found in winter when the ground is usually warmer than the surface, yielding to a positive contribution (Fig. 3a and 4c). The advection term does not present a strong monthly-hourly cycle compared to the other terms, although one can distinguish a positive action (still very low) to local temperature variations at

all seasons except winter during daytime, as shown in Figure 3a and 4e.

Lastly, the residual, defined as the difference between the model (sum of all terms) and the observations (i.e. $\frac{\partial T_{2m}}{\partial t}_{mod} -$

$\frac{\partial T_{2m}}{\partial t}_{obs}$), also has a seasonal variability and is mainly negative, but a minimum difference of -0.45 °C h⁻¹ between the two datasets is found in April around noon, which is related to the negative increase of the HA term at that time. This is due to the important absence of latent heat flux data especially for this month, which implies an increase in the bias when $\tau_a$ is estimated

(see Table 1 and Appendix A). Nevertheless, during the months when solar radiation is strong, the residual reaches positive values (with a maximum value around 0.4 °C h⁻¹) but remains a low overestimation. These values for the residual are low compared to the magnitudes of $R_{CS}$, but it remains almost at the same order of magnitude as the $R_{CL}$ and HA terms and a good correlation in a diurnal cycle approach is found. Generally, this hourly mean residual could also be due to the simplifications and assumptions made in the model, under-sampling and energy imbalance.

**4 Weight of the different terms acting on temperature variations**

In this section, a random forest evaluation (James et al., 2013; Manish, 2016; Brownlee, 2016; Loh, 2002) is carried out at different time-scales to estimate the relative weight (i.e. importance) of each term in temperature changes according to the hour, month or seasons.

**4.1 General behavior**

Here, the random forest method is used to establish which term dominates the temperature changes in the model ($\frac{\partial T_{2m}}{\partial t}_{mod}$) during day and nighttime. This machine learning method consists of bootstrapped-aggregated decision trees, a method that combines and gathers together the results of these trees to construct more powerful prediction models. Since in the present study the model has already been constructed, one of its most impressive features is used, which consists of the ability to provide a fully nonparametric estimation of the importance of each term (or predictor) on the model. One of the main

advantages of this method is that it allows covering not only the impact of each term individually in the model but also the





multivariate interactions with other predictors. Indeed, other approaches to estimation of predictor importance (e.g. simple squared marginal correlations method, squared standardized coefficients) do not give reliable results when the problem involves correlated predictors (Grömping, 2007). Additionally, this method is used as a result of the nonlinear relationship between each separated term and the model (not shown), which does not yield to an estimation and quantification on how each

process at the surface affects the temperature variations, an approach suggested and used by Miller et al. (2017) who estimated the response and importance of some surface processes (such as $F_{lat}$ and $F_{sens}$) to the forcing radiative terms in Summit, Greenland.

The predictor importance estimate value is defined as the sum of the mean square error (MSE) of each term averaged over all decision trees used and normalized by the standard deviation taken over the trees. This principle consists of permuting the

values of a term – predictor – in the decision trees where this term was left out-of-bag and assess how much worse the MSE becomes after the permutation (James et al. 2013, Chapter 8). Thus, the larger this value, the more important the term. This importance estimation feature from the random forest method has been previously used to calculate the variable importance in different datasets for many applied problems in sciences or health fields for both regression and classification studies (e.g. Archer et Kimes 2008; Genuer, Poggi, et Tuleau-Malot 2012; Strobl et al. 2007).

Figure 5a and b present the predictor (i.e. each term involved in $\frac{\partial T_{2m}}{\partial t}_{mod}$, see Eq. (2)) importance estimate value for daytime and nighttime respectively. Figure 5a corroborates that $R_{CS}$ is the most important term, followed by $R_{CL}$ and then HA, whereas HG is the least important term in the model developed for the timescale considered. Next, Figure 5b illustrates that $R_{CL}$ dominates hourly temperature variations during nighttime, followed by HG and $R_{CS}$. These results agree with Kukla and Karl (1993), who suggested that the key regulator of nighttime warming temperatures is downward thermal radiation when cloudy

cases are presented, an effect that is damped most of the time by the radiative cooling effect of the ground.

**4.2 Diurnal cycle of the weights**

The separation of daytime/nighttime does not allow the consideration of the importance of processes occurring during transition times of the diurnal cycle when the increase/decrease in temperature is the strongest. So, to better evaluate the influence of each term on hourly temperature variations at different times of the day, the importance estimation value is

determined for each hour using the random forest method described previously in Section 4.1.

Figure 6 presents the results of this method for each season. As expected (and previously exposed), for all the seasons $R_{CL}$ is the term dominating during nighttime, just after sunset, and before sunrise (indicated by vertical dashed lines in black).

After sunrise, the surface heating produced by the sun in the early morning enhances a high growth rate in the temperature variations, whose effect makes $R_{CS}$ the dominant term driving $\frac{\partial T_{2m}}{\partial t}_{mod}$ at those hours of the day for all seasons, except for

winter. For this latter season (Figure 6a), the growth rate of $R_{CS}$ is almost the same as that of $R_{CL}$ and thus it does not expose an important estimation value. This effect is due to the weak mean solar zenith angle (SZA) for this season, and the surface heating by the sun in clear-sky conditions is not strong enough to modulate temperature variations. On the contrary, $R_{CL}$


importance reaches its minimum just after sunrise and before sunset, where, depending on the season, either $R_{CS}$ or HA are the main terms controlling temperature variations.

A shift in importance between $R_{CL}$ and $R_{CS}$ occurs during the rest of the day for the four seasons of the year. This effect is explained by the variation of the data for these two terms: $R_{CS}$ standard deviation for a given hour/season is weak, and thus its influence to explain the difference between one day to another at a specific hour remains minimal and it is not a strong predictor at diurnal cycle scale, especially in the summer and spring (Figure 6b and c, respectively) when other variables will modulate temperature variations. On the other hand, $R_{CL}$ turns into the main modulator of $\frac{\partial T_{2m}}{\partial t}_{mod}$ for all the seasons due to its strong

standard deviation for a given hour/season, reaching its maximum importance in summer (Figure 6c). Therefore, the hourly temperature variations are more sensitive to cloud changes rather than solar radiation which does not vary significantly for a specific hour from day to day.

Concerning the other terms, HA importance grows along the day. Its variation is greater than that of the other terms making it the second most important modulator most of the time (except for autumn, Figure 6d). Its contribution is weak in winter due

to the lack of solar radiation and vegetation whose absence will diminish the turbulent heat fluxes measured at the surface, along with a weak MLD. Note that it becomes sometimes the most important term in the late afternoon just before sunset. This is explained by an increase in instability of the ABL enhanced by a large boundary layer and surface temperature gradient (i.e. the difference between $T_{MLD} - T_{2m}$ reaches its maximum, see the third term of the right-side of Eq. (7)) in the late afternoon (not shown) which is generated by the increase in turbulence fluxes near the surface.

Concerning the Adv term, it does not show a significant contribution during daytime but it becomes the second most important modulator during nighttime before sunrise for all seasons except summer; probably a thermally-induced circulation linked to the urban heat island which is set up in Paris is at the origin of this importance: this circulation is likely to create a greater variation of temperature some nights when cooler air from rural areas is advected to warmer zones towards the city center. The SIRTA observatory, located in a suburban area at 20 km from the center of Paris could be indeed regularly affected by this

modulation of circulation.

Figure 6 supports a clear and reliable estimation of the importance of each term split into seasons for every hour of the day, which is not seen by estimating the diurnal and annual cycle contribution of each term to temperature variations (cf. Section 3). Indeed, depending on the hour of the day (and thus the state of the atmosphere), one term will become important over the rest and temperature variations will be more sensitive to its change even if its contribution in terms of magnitude remains

smaller compared to other terms. For instance, $R_{CS}$ absolute contribution during nighttime is higher than that due to $R_{CL}$ (Fig. 3b), and yet the latter is more important during nighttime due to its seasonal and hourly (not shown) contribution variability to $\frac{\partial T_{2m}}{\partial t}_{obs}$.





## 5 Discussion on the specific role of clouds on temperature variations

Section 4 shows that clouds are the main modulator of solar radiation on hourly temperature variations during the day (and the
main contributor during the night). Knowing how each term affects the temperature variations and when they offset each other,
a deeper analysis is carried out in this section, to understand the role of clouds. This analysis is performed by considering other
variables available in the SIRTA-ReOBS dataset to characterize both the atmosphere and the clouds.

In the following, only cloudy cases are considered. These cases are identified based on a criterion on the absolute value of
CRE that must be higher than 5 W m$^{-2}$ in SW and LW (Chiriaco et al., 2018) during daytime, whereas for nighttime only LW
is considered for this criterion. The SWCRE (and LWCRE) is calculating following Eq. (11):

$$SWCRE = F_{SW}^{\downarrow} - F_{SW,CS}^{\downarrow} \tag{11}$$

According to this threshold, cloudy conditions correspond to 82% of the total cases from January 2009 to February 2014, and
unique-clear-sky conditions represent the remaining 18%.

### 5.1 Daytime analysis

Since the $R_{CS}$ term dominates during daytime, the ratio of $\frac{\partial T_{2m}}{\partial t}_{obs}$ divided by $R_{CS}$ is created to estimate how much these two
terms are driven by the solar radiation: when this ratio is close to 1, it means that the variations of temperature are the ones
that would be expected in clear-sky conditions with no other modulators, and when this ratio deviates from 1 it means that the
temperature variations are damped or enhanced by the other terms. The $R_{CL}/R_{CS}$ ratio is also estimated, of which the
distribution (not shown) has two peaks, one between -0.5 and -1 and one slightly negative with a tail in positive values. Thus,
three bins are created from the highest cooling effect to the warming effect: (i) $-1.0 \leq \frac{R_{CL}}{R_{CS}} < -0.5$ (bin 1), (ii) $-0.5 \leq \frac{R_{CL}}{R_{CS}} <$
0 (bin 2) and (iii) $0 < \frac{R_{CL}}{R_{CS}} \leq 0.5$ (bin 3). The cases with negatives or very close-to-zero values of $R_{CS}$ (10% of the cases),
which occurred in early mornings and late afternoons, are excluded since they affect the sign of the distribution or give
divergent values.

Figure 7a shows the values of the observed temperature variations divided by $R_{CS}$ and Figure 8 shows the distribution
360   (represented as box-and-whiskers) of different relevant meteorological observations available in the SIRTA-ReOBS dataset,
for each bin created, split into seasons for daytime. To add cloud information, lidar data are also considered but because they
are not available all the time, and in particular when it is raining (see Sect. 2.2), they are presented as additional boxplots (light
color ones) that correspond to the lidar sampling (whereas the dark color boxplots are for the total cloudy sampling). This
difference of sampling mostly affects the first two bins when clouds have a cooling effect, for which both the occurrence and
365   the amount of precipitations are the highest (not shown). Lidar SR(z) histograms presented in Figure 9 are estimated by
cumulating all lidar SR(z) observations available for one bin and one season in particular. The red horizontal lines on each
histogram correspond to low- ($P > 680$ hPa), mid- ($440 < P < 680$ hPa), and high-level ($P \geq 440$ hPa) clouds limits (Chepfer





et al. 2010; Chiriaco et al. 2018). Note that a Noise bin is presented here, which is simply the sum of the -999 and -777 bins that correspond to noisy and non-normalized profiles (cf. Section 2.2).

### 5.1.1 Case with strong cloud cooling effect - bin 1

Figure 7a shows that the values of temperature variations (normalized by $R_{CS}$) are the lowest for clouds having the most cooling effect (bin 1 in blue) for all the seasons. In winter this ratio stays mostly positive (temperature increase in one hour) but less than 0.5 (less than half the increase that could be reached if the sky was clear) and it can become negative during summer and fall, i.e. the warming induced by $R_{CS}$ can be counterbalanced by the clouds in these seasons. Besides, the cloud cover is almost total and the radiative effects are strong, as seen in Figures 8d, e and f. In addition, the seasonal variability of SWCRE is directly related to the seasonal variability of $F_{SW,CS}^{\downarrow}$, and not (or only slightly) to the seasonal variability of cloud properties, since the ratio $SWCRE/F_{SW,CS}^{\downarrow}$ remains almost constant during the entire year (not shown) and cloud cover presents the highest mean values (Fig. 8d).

The light blue bin is the same as the dark one except for lidar sampling, i.e. for non-rainy cases exclusively. Indeed, lower RH values (Fig. 8b) favor lower precipitation occurrence (not shown), higher amount of $F_{sens}$ (Fig. 8c), and higher (i.e. less negative) values of SW CRE (Fig. 8e) for the lidar sampling compared to the total one. Logically, lower LWCRE is found for the lidar sampling. As expected, temperatures are higher for the lidar sampling for all the seasons except summer, as shown in Fig. 8a, along with a more frequent negative temperature variation (Fig. 7a). For this sampling that excludes rainy cases, both higher and lower values of SWCRE are found for spring rather than for summer (Fig. 8e): the minimal value is around $\sim -600$ W m$^{-2}$ for spring whereas for summer it is $\sim -500$ W m$^{-2}$ (a situation not captured by the total sampling where summer has a lower value of SWCRE than spring). Figure 9b shows that low- and high-level clouds are more frequent for spring than for summer (Fig. 9c) and thus stronger negative values of SWCRE are found for spring. Then, the important presence of mid-level clouds with a high value of lidar SR(z) (> 80) spotted in summer in Fig. 9c are potentially at the origin of the strong negative values of SWCRE despite the very low presence of other clouds. In addition, despite the strong negative values of SWCRE for this bin for all seasons, surface temperatures are not so low (Fig. 8a), partly due to the high LWCRE values (Fig. 8f) that slightly dampen the strong SWCRE.

Some differences in cloud presence can be detected in Figures 9a-d for the first bin. For the fall season, the SR(z) histogram in Fig. 9c exhibits an important presence of high-level thin clouds (especially above 8 km), along with mid-level thick clouds. Figure 9b also presents an important presence of high-level clouds in spring but within a smaller vertical range (7 < z < 10 km) compared to fall. Indeed, these high-level clouds could correspond to cirrostratus or cirrocumulus which form when a mass of warm air meets a mass of cold air, where the lighter warm air rises and could form these cirrus clouds, an event that occurs more often in the transition seasons, such as spring and fall. In addition, spring exhibits high amounts of low-level thick clouds (SR>40) that are not detected the rest of the year, which could correspond to the moments just before a storm is set. Indeed,



the highest amount of precipitation rates are found for this season (not shown) and the cloud cover minima in Fig. 8d for the
lidar sample is 90% (the highest among the other seasons) and less clear-sky conditions are found ($0.01 < SR(z) < 1.2$).

### 5.1.2 Cases with weak cloud radiative effect: cooling or warming

When $R_{CS}$ becomes dominant with respect to $R_{CL}$ (bins 2 and 3), temperature variations are most of the time positive,
especially in winter (Fig. 7a). In comparison with the first bin, the air becomes drier (higher sensitive heat flux and lower RH,
Figure 8(b) and (c)), which agrees with less cloud cover (Fig. 8d), and lower LWCRE and less negative SWCRE (Fig. 8e and
f). Figure 9 shows fewer differences between bin 1 and bin 2 ($R_{CL} < 0$) than between bin 2 and bin 3 ($R_{CL} > 0$) because this
figure only considers non-rainy situations and in section 5.1.1. it is noted the important differences between the two samplings
(total and non-rainy) for bin 1 for most of the variables. This difference between the two samplings is weaker for bins 2 and 3,
despite its existence in winter and fall. Figure 9 is thus more representative of the full sampling for these two bins than it is for
the first bin.
For the lidar sampling, LWCRE in winter and fall is slightly higher than for the two other seasons for the second bin (Fig. 8f).
This is partly due to the presence of high-level thin clouds ($5 > SR(z) > 20$), such as cirrus, detected in the histograms of $SR(z)$
for this bin in Fig. 9e and h for these two seasons, which is slightly more than for spring and summer.
In bin 3, the absence of mid-level clouds and the lower occurrence of high clouds compared to bin 2 is obvious, while the
difference for low-level clouds is not clear. This is consistent with lower LWCRE (Fig. 8f) but not necessarily with the very
low values (i.e. close to zero) of SWCRE observed for bin 3 (Fig. 8e). The SWCRE values close to zero are explained by the
fact that most of the hours corresponding to this bin 3 coincide just after sunrise (not shown), when solar radiation is still weak,
thus explaining that $F_{SW}^{\downarrow} \approx F_{SW,CS}^{\downarrow}$. These low SW and LWCRE values for the lidar sampling also correspond to low mean
cloud cover (Fig. 8d) and thus a lower amount of clouds are detected by the $SR(z)$ histograms (Figure 9i-l). Note that some of
the SWCRE values are even positive, revealing that $F_{SW}^{\downarrow} > F_{SW,CS}^{\downarrow}$, an effect observed when the direct part of solar radiation
is not fully attenuated and when some diffuse radiation reaches the surface in addition to the direct flux, radiation which is
strengthened by scattering processes in clouds and atmospheric particles. This effect thus increases observed radiation beyond
a respective clear-sky scenario, and it mostly happens in winter (but could occur the rest of the year) in early mornings and/or
late afternoons hours when the solar elevation angle is weak (Stapf et al., 2020; Wendisch et al., 2019). This distribution
towards early morning hours explains that despite the high values of the ratio between temperature variations and $R_{CS}$ (Fig.
7a) and lower cloud cover and cloud effects, surface temperatures are not always maximal for this bin (these maximal
temperatures happen only for spring and fall).

### 5.2 Nighttime analysis

During nighttime, a similar procedure is followed to analyze and characterize different meteorological variables under different
cloudy conditions. Since $R_{CL}$ is the term dominating and controlling temperature variations (Section 4) and since $R_{CS}$ is





constant, it is not necessary to divide it by $R_{CS}$. Thus, two bins are created from the PDF of $R_{CL}$: $0 < R_{CL} < 0.75 \,°\text{C h}^{-1}$ and

$0.75 \leq R_{CL} \leq 1.5 \,°\text{C h}^{-1}$. During nighttime there is no SW radiation so clouds always have a warming effect on surface

temperature. The distribution of $\frac{\partial T_{2m}}{\partial t}\Big|_{obs}$ for the two bins created for all the seasons is presented in Fig. 7b. Figure 10 shows

the same meteorological variables shown in Fig. 8 but for nighttime cases. Cloud fraction profiles are not shown because the

lidar does not operate during nighttime.

Figure 7b shows that the temperature variations are generally stronger for the first bin than for the second bin, i.e. the negative

temperature variations induced by longwave cooling during nighttime are damped when the effect of clouds is stronger. This

agrees with what is expected since, at hourly time scales, clouds are the most important factor in modulation of temperature

variations during nighttime (see section 4). As expected, the second bin has stronger LWCRE values (Fig. 10e). The effect of

clouds seems enhanced when cool and moist air is present over SIRTA (Fig. 10a, b). More details for each bin are given in the

following subsection.

**5.2.1 Case with weak cloud warming effect**

For the first bin ($0 < R_{CL} < 0.75 \,°\text{C h}^{-1}$), LWCRE presents a wide range of values going from 5 W m$^{-2}$ up to 75 W m$^{-2}$ with

no seasonal variability detected (Fig. 10d), and yet temperatures are the highest for this bin (Fig. 10a) except in winter. As

expected, upward sensible heat flux is most of the time negative due to surface cooling, as values of $\frac{\partial T_{2m}}{\partial t}\Big|_{obs}$ are predominantly

negative (Fig. 10c and 7b, respectively), with drier air conditions since RH has the lowest values for all the seasons (Fig. 10b).

**5.2.2 Case with strong cloud warming effect**

The temperature distribution has the lowest values for all seasons (except spring) for the category of clouds that have the

strongest warming effects (second bin, in purple on Fig. 10a). This category of clouds is tightly linked to strong LWCRE

ranging from 55 W m$^{-2}$ up to 90 W m$^{-2}$ (Fig. 10d). These low temperatures could be associated with situations when the

incoming air masses are colder, which is linked to large-scale situations (Pinardi and Masetti, 2000; Wang et al., 2005). Note

that here hourly temperature variations are studied, and indeed on multi-day time scales, it is the large-scale atmospheric

processes that determine on the first order the daily temperature value.

Finally, the distribution of sensible heat flux is positive in winter with values reaching up to 40 W m$^{-2}$ (purple box plot, Fig.

10c). These unusual and positives values during nighttime are explained by the presence of a strong stable atmospheric layer,

where surface temperatures are the lowest except for spring (Fig. 10a) and the atmosphere is warmer at higher altitudes since

clouds contribute to warming the most at this time, and therefore a positive upward sensible heat flux develops. Similar

behavior was previously found by Miao et al. (2012) in Beijing also during nighttime in cloudy conditions, where sensible and

latent heat fluxes were close to zero due to the presence of a stable atmospheric layer, especially in winter when the sensible

heat flux is found to be slightly greater than the latent heat flux. Indeed, almost half of temperature variation values are found

to be positive for this season, as illustrated in Fig. 7b, and along with the positive upward sensible heat flux found, higher



temperatures are found for this bin which is not seen in the other seasons where the sensible heat flux is close to zero or mostly negative and temperatures are higher for the first bin (cloud with a low warming effect).

## 6 Conclusions

The European climate and most temperature anomalies are affected not only by the circulation of large-scale air masses (such as the NAO or blocking regimes) but also by smaller-scale processes such as cloud radiation and surface-atmosphere interactions located within the atmospheric boundary layer. In this paper, a method is developed and evaluated to quantify each process that affects hourly 2 m-temperature variations on a local scale (near Paris), based almost exclusively on observations. The method exhibits good accuracy and is able to quantify well the realistic diurnal and monthly-hourly cycles of each term involved, especially in summer where statistics are the best. The clear-sky radiative term represents the biggest positive (negative) contribution during daytime (nighttime) with values reaching up to $4 \, ° \, h^{-1}$ ($-1.7 \, ° \, h^{-1}$), whereas clouds cool (warm) up to $-3.7 \, ° \, h^{-1}$ ($1.4 \, ° \, h^{-1}$). The atmospheric heat exchange becomes predominant in the late afternoons when turbulent fluxes develop due to the increase in atmospheric instability. Some of the biases found are due to the difficulty of the model to well reproduce smaller-scale processes such as the cold pool events.

The use of a random forest analysis makes it possible to identify which term dominates over the others. Clear-sky radiation most influences the 1-hour temperature variations during the day, whereas it is the cloud radiation during the night followed by the ground heat exchange term. Nevertheless, separating the dataset into hours and seasons shows that the cloud radiation effect becomes predominant in several hours for all the seasons because it presents a greater variation of its hourly values than the clear sky radiative effect, especially in spring and summer during daytime when more local cloud cover variability occurs. This cloud dominance is still found despite the fact that the clear-sky radiation magnitudes are the highest on a monthly-hourly scale. Temperature variation is then more sensitive to cloud changes within an hour rather than the large contribution of clear-sky radiation which mostly dominates the early mornings when surface heat fluxes are not yet developed. The other terms remain important as there are times when some of them modulate temperature variations, especially the turbulent heat fluxes in the late afternoons. These terms also remain necessary to obtain the best coefficient estimator between the directly measured observations and the method developed.

To better understand the importance of clouds on temperature variations at this scale, a deeper analysis is performed partly based on lidar profile observations. The atmosphere presents a high amount of cloud fraction detected by both the lidar and the sky imager during daytime which correlates well with the strong cooling contribution found for the cases when clouds have a strong cooling effect. Indeed, an important presence of mid-level thin and thick clouds is spotted for all the seasons (except spring) for these cases. These thick clouds could correspond to nimbostratus which are very opaque and often produce continuous moderate rain, but the lidar detects them just before the rain starts. Indeed, temperatures at this time are very low, which contributes to these clouds forming since they usually form ahead of a warm front. On the contrary, a weak cooling effect is predominantly associated with low- and high-level thin clouds for all seasons. Situations with daytime positive cloud





radiative effect occur when the atmosphere is close to clear-sky conditions and corresponds to low- and high-level thin clouds mostly in early mornings and late afternoons. In addition, situations with weak cloud effect (either negative or positive) are

associated with an important presence of high-level thick clouds for all the seasons (except winter), whose LWCRE is strong but SW clear-sky radiation dominates and controls temperature variations along with other surface meteorological variables, high-level clouds which are more present than the times of strong cooling effect. Overall, a dominance of mid- and high-level clouds at the SIRTA observatory is detected for all cases. The dominant presence of this specific type of cloud is linked to the geographical position of the SIRTA observatory since it is located in a zone where warm and wet air coming from the Atlantic

Ocean can nudge against the cold and dry masses of air coming from the Siberian region. This encounter of air masses (whether it is the warm air that encounters cold air or the opposite) tends to form mostly high-level clouds. Similar behavior has been found by Chakroun et al. (2018) during nighttime and Mariotti et al. (2015) over the Euro-Mediterranean area where the variability of cloud fraction (CF) is driven mostly by the encounter of two different air masses coming from Northern Europe and Southern Africa. During nighttime, even when clouds have a positive important radiative effect, temperatures are low and

they are more controlled by large-scale processes since surface turbulent fluxes are most of the time low and there is no shortwave radiation. Since neither cloud fraction variable nor lidar data are available during nighttime, no information about the percentage of cloud presence can be known, yet variability on LWCRE is still found between clouds whose warming effect is weak and those whose warming effect is strong. This LWCRE variation is on the first-order controlled by the presence of clouds, but it could also be due to the sensitivity of LW radiation arriving at the surface to the presence of atmospheric gases

and aerosols and not only to cloud radiative properties, as shown previously by Dufresne et al. (2002); Satheesh and Krishna Moorthy (2005); Kushta et al. (2014). Besides, the strong stability of the atmosphere at night creates positive sensible heat flux when clouds have the warmest effect on temperature variations.

The approach developed in this study is innovative because it is predominantly based on observations. It opens several perspectives on the possibility for continuing work: (i) use of the approach in combination with weather regimes to better relate

the small-scale processes to the large-scale atmospheric dynamics; (ii) application of the approach to other locations to understand the spatial variability of the results and specific local conditions affect each of the terms involved in surface temperature variations; (iii) identification of the moments when one term becomes predominant over the rest by integrating the time into larger scales (such as weekly and monthly); and (iv) estimation of how well these estimations are represented in the present and future climate simulations to evaluate models and/or understand the evolution of the different terms in a

warming climate. Further, the ground heat exchange contribution is very site-dependent (more than the other surface terms), with different behavior in an urban environment, therefore it will be interesting to study its impact in the attenuation/amplification of maximum surface temperatures in periods of heatwaves.

*Author contributions.* OR carried out the data analysis, methodology and prepared all the figures. MC, SB and JR contributed

to the data analysis, advised on methods, validation and interpretation of results. OR wrote the manuscript with contributions from MC and SB.



*Acknowledgements.* The authors would like to acknowledge Mesocentre ESPRI team at IPSL for providing storage resources and giving access to SIRTA Re-OBS dataset, and also the SIRTA observatory for collecting and providing the geophysical

variables and lidar data used in this study. The advection term as well as the temperature at the mixing layer depth were estimated using Copernicus Climate Change Service Information ERA5 [2009-2014].

*Competing interest.* The authors declare that they have no conflict of interest.

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






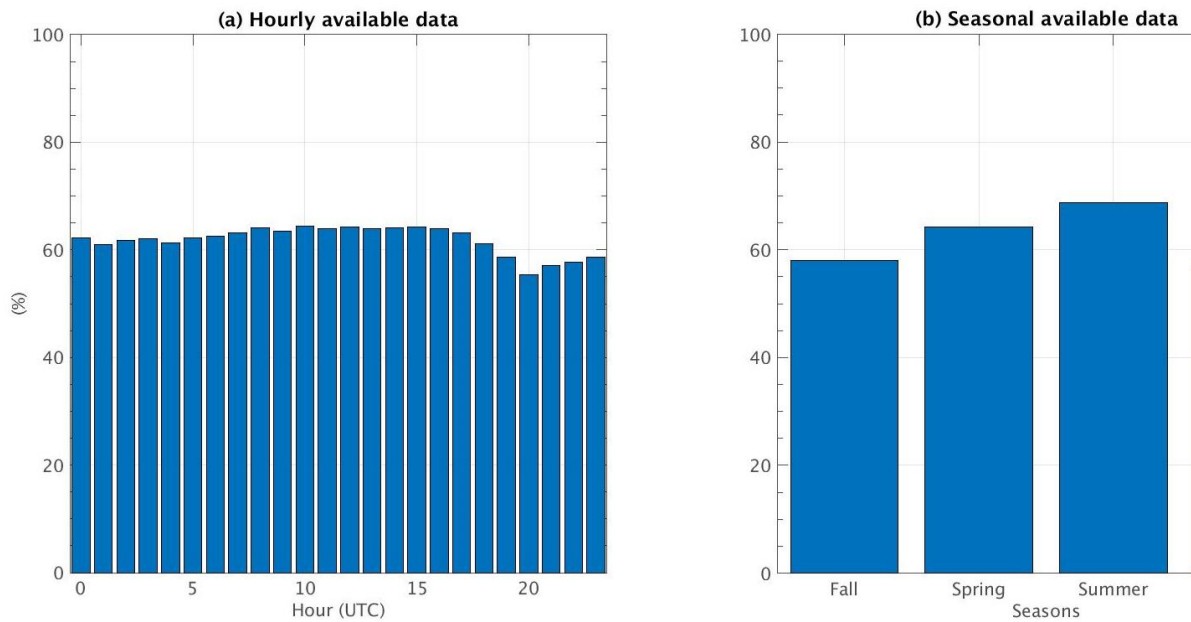

**Figure 1: Histograms of (a) hourly and (b) seasonal available data in the SIRTA ReOBS dataset, considering all the variables needed for the period of study 2009-2014 for the temperature variation model.**






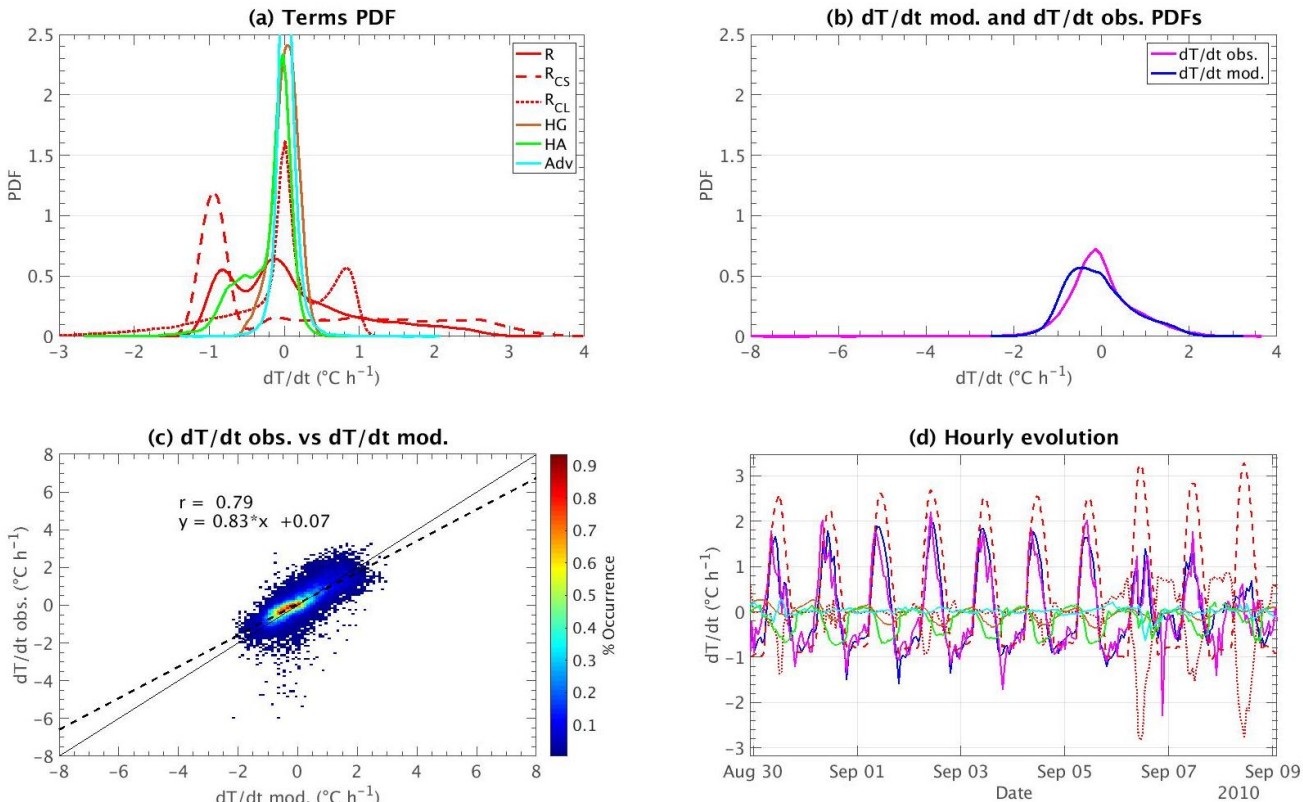

**Figure 2: PDFs of (a) radiation (red line), radiative clear sky (red dashed line), radiative cloud (red dotted line), heat ground (brown line) and heat atmospheric (green line) exchange, and advection (cyan line) terms, and (b)** $\frac{\partial T_{2m}}{\partial t}_{obs}$ **(pink line) and the** $\frac{\partial T_{2m}}{\partial t}_{mod}$ **(blue line). (c) is the scatter plot of** $\frac{\partial T_{2m}}{\partial t}_{obs}$ **vs** $\frac{\partial T_{2m}}{\partial t}_{mod}$**. The sloping solid line represents the 1:1 line and the dashed one correspond to the least-square best fit linear regression line between the two datasets. The correlation coefficient "r" and the linear equation fitting the best for the two datasets are also indicated. (d) Hourly evolution for September 2010 of the temperature variation for the different terms, the observations, and the model (same colors as in (a) and (b)).**





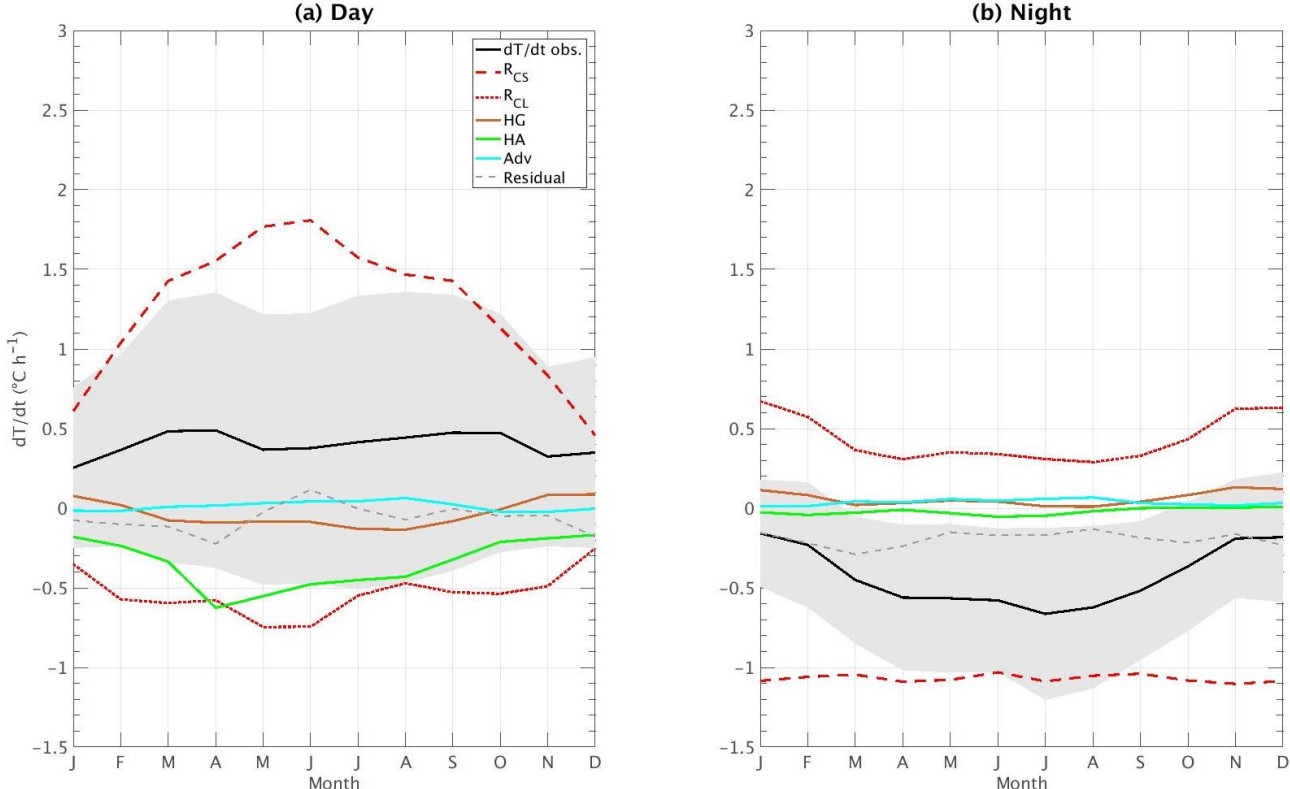

**Figure 3: Mean annual cycle averaged monthly from 2009 to 2014 of the five terms of the model, the observations, and the residual (dashed gray lines) for (a) day and (b) night. Same colors as in Fig. 2a and b. The shaded gray area in each of the subfigures represents the standard deviation of $\frac{\partial T_{2m}}{\partial t}_{obs}$.**





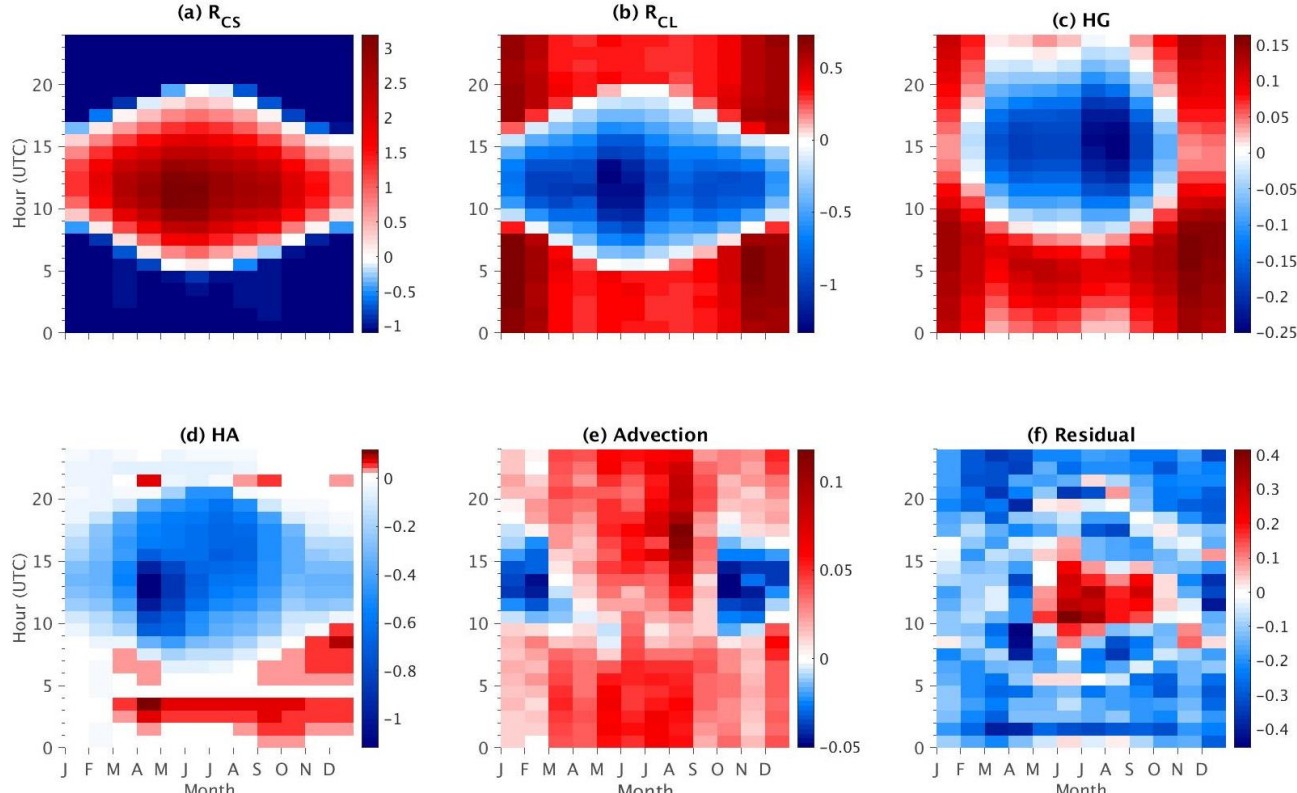

**Figure 4: Monthly-hourly mean values for (a) $R_{CS}$, (b) $R_{CL}$, (c) HG, (d) HA, (e) Adv and (f) the residual (i.e. difference between the model and the observations). Units on the color bars are all in °C h$^{-1}$, and their scale is different for each subfigure.**





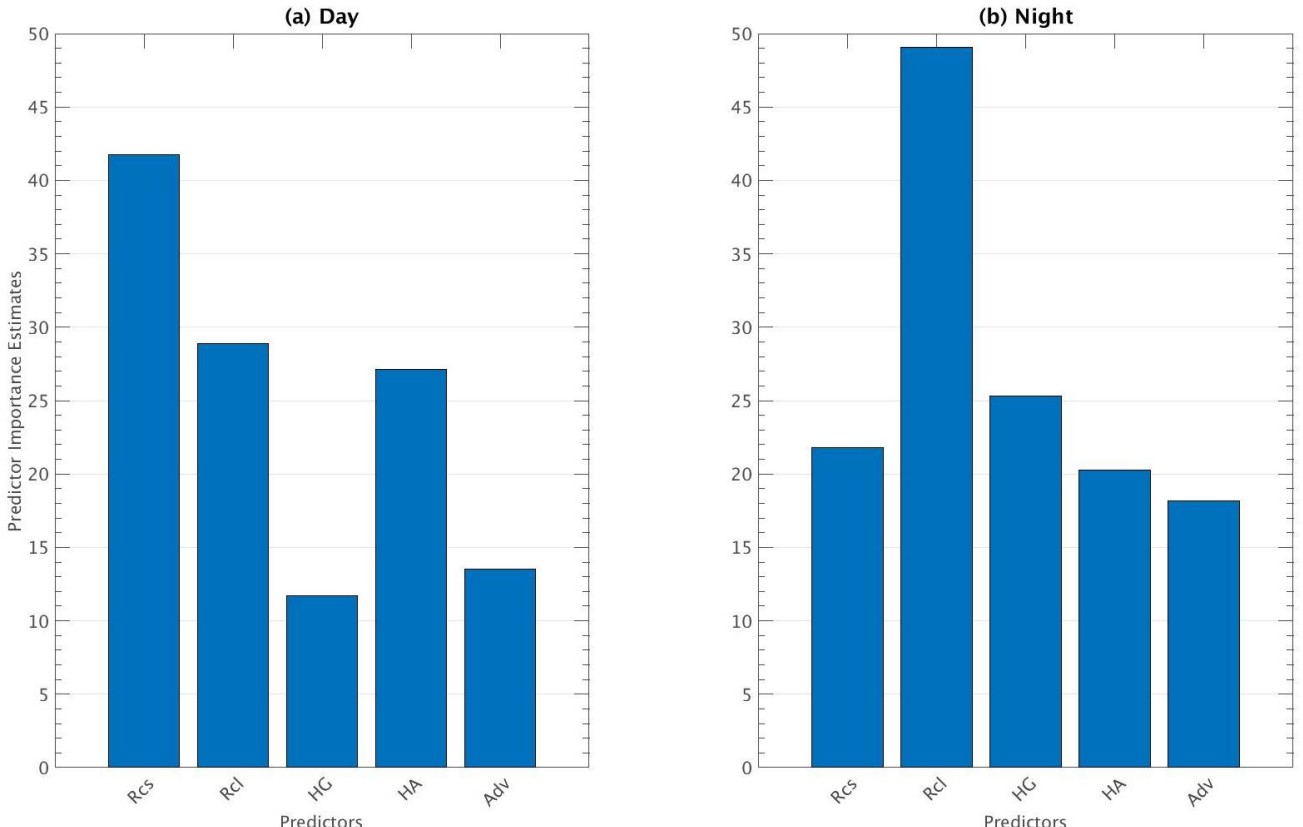

**Figure 5: Predictor importance estimates obtained by the random forest method for (a) day and (b) night. The abscissa in both cases represents each predictor (or term) of the model, and the y-axis their importance (unitless) defined as the sum of their mean square error when permutation in the decision trees is done.**






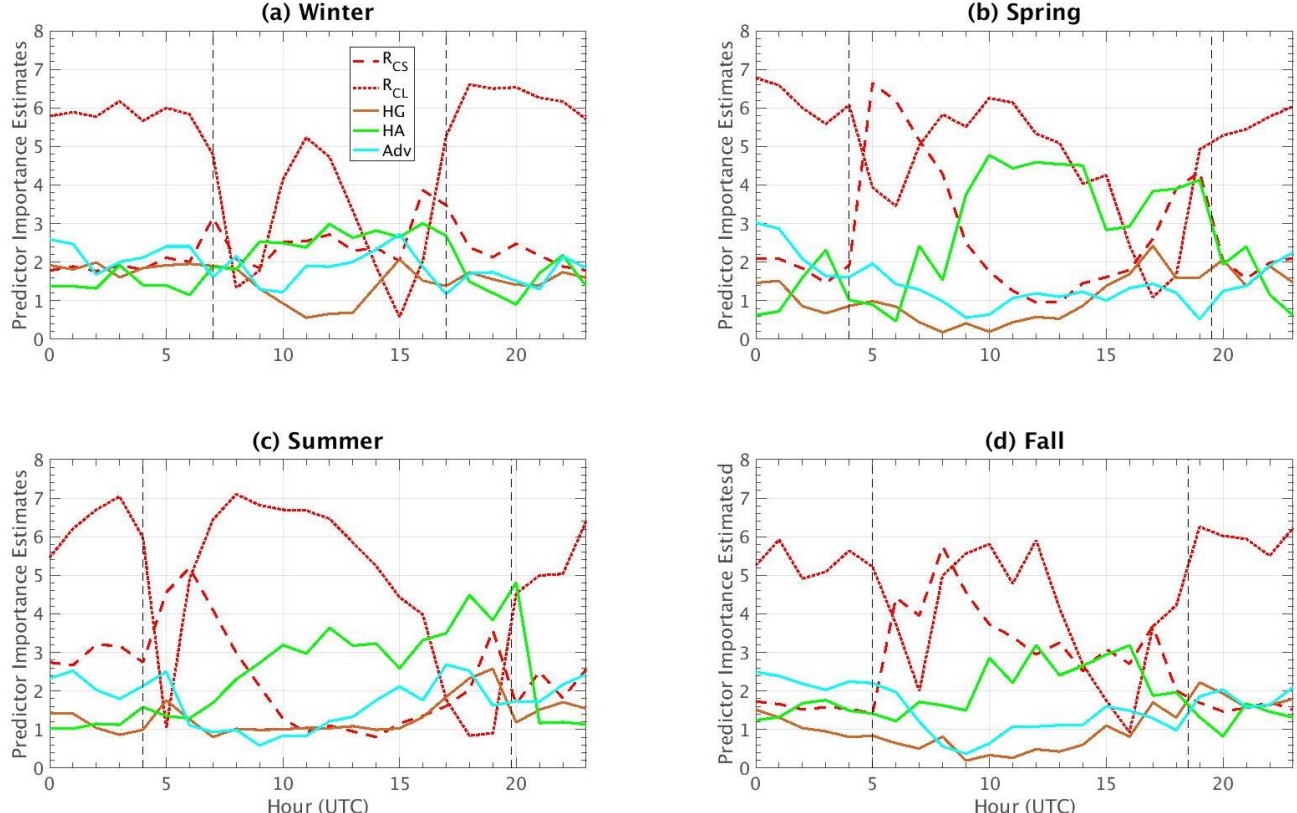

**Figure 6: Diurnal cycle of the predictor importance estimate for each term of the model, for (a) winter, (b) spring, (c) summer and (d) fall. Same colors as in Fig. 2a and b. Vertical dashed black lines are for sunrise and sunset mean hours.**




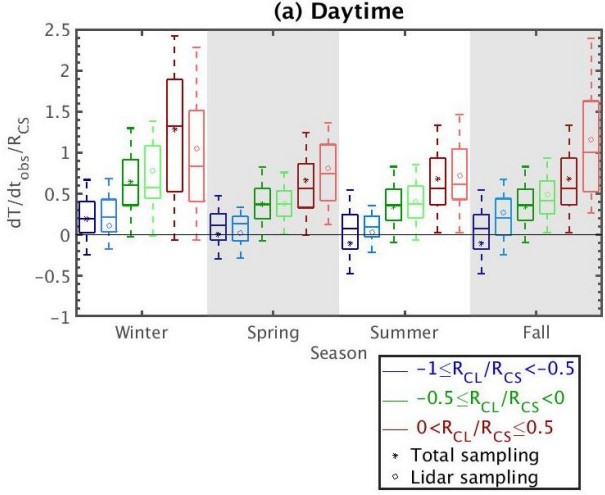

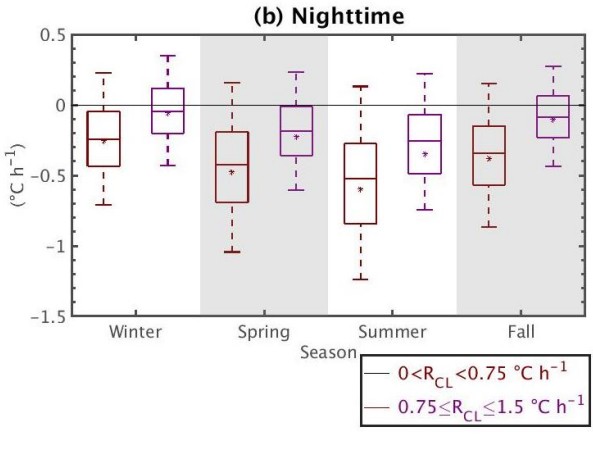


**Figure 7: (a) (a)** $\frac{\partial T_{2m}/\partial t_{obs}}{R_{CS}}$ **daytime values for the four seasons for three** $R_{CL}/R_{CS}$ **bins:** $-1.0 \leq \frac{R_{CL}}{R_{CS}} < -0.5$ **in blue,** $-0.5 \leq \frac{R_{CL}}{R_{CS}} < 0$
**in green and** $0 < \frac{R_{CL}}{R_{CS}} \leq 0.5$ **in red. Dark-colors box plots with the mean represented as '\*' correspond to cases when all**
**meteorological variables are available at the same time (without considering lidar availability), whereas light-colors box plots with**
**'o' as their mean value represent the sample when both all meteorological variables and Lidar profiles are available simultaneously.**

**(b)** $\partial T_{2m}/\partial t_{obs}$ **nighttime values for two** $R_{CL}$ **bins:** $0° h^{-1} < R_{CL} < 0.75° h^{-1}$ **in maroon and** $0.75 \leq R_{CL} \leq 1.5 °C h^{-1}$ **in purple.**
**Distributions are represented by box-and-whiskers plots, where the boxes indicate the 25th and the 75th data percentiles, the whiskers**
**indicate 5th and 95th percentiles, the middle line represents the median, and the \* or º is the mean. Negative and close-to-zero values**
**are removed (see text for further information).**





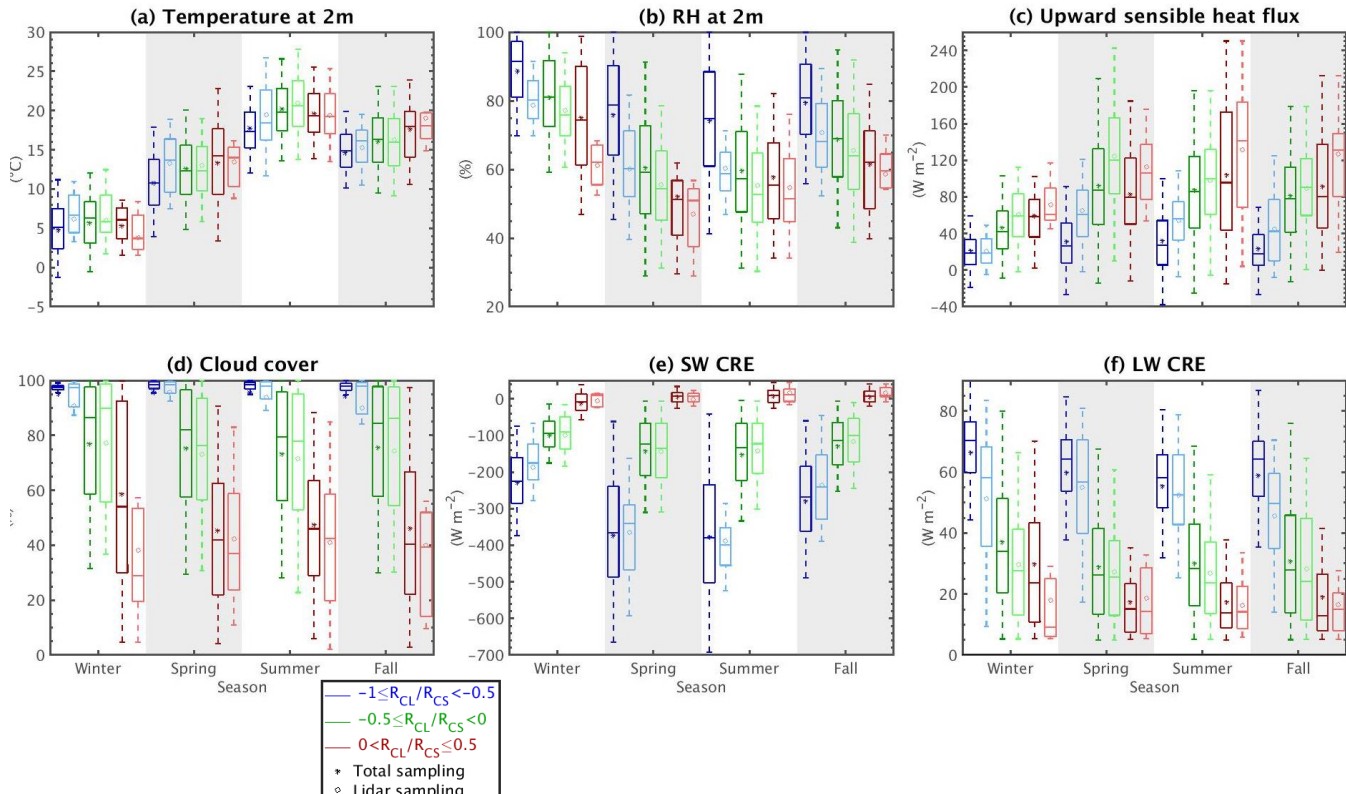


**Figure 8: Daytime values for (a) temperature, (b) relative humidity, (c) upward sensible heat flux at 2m, (d) cloud cover retrieved from a sky imager, (e) shortwave and (f) longwave cloud radiative effect. Colors and boxplots follow the same definition as in Fig.**
**7a.**





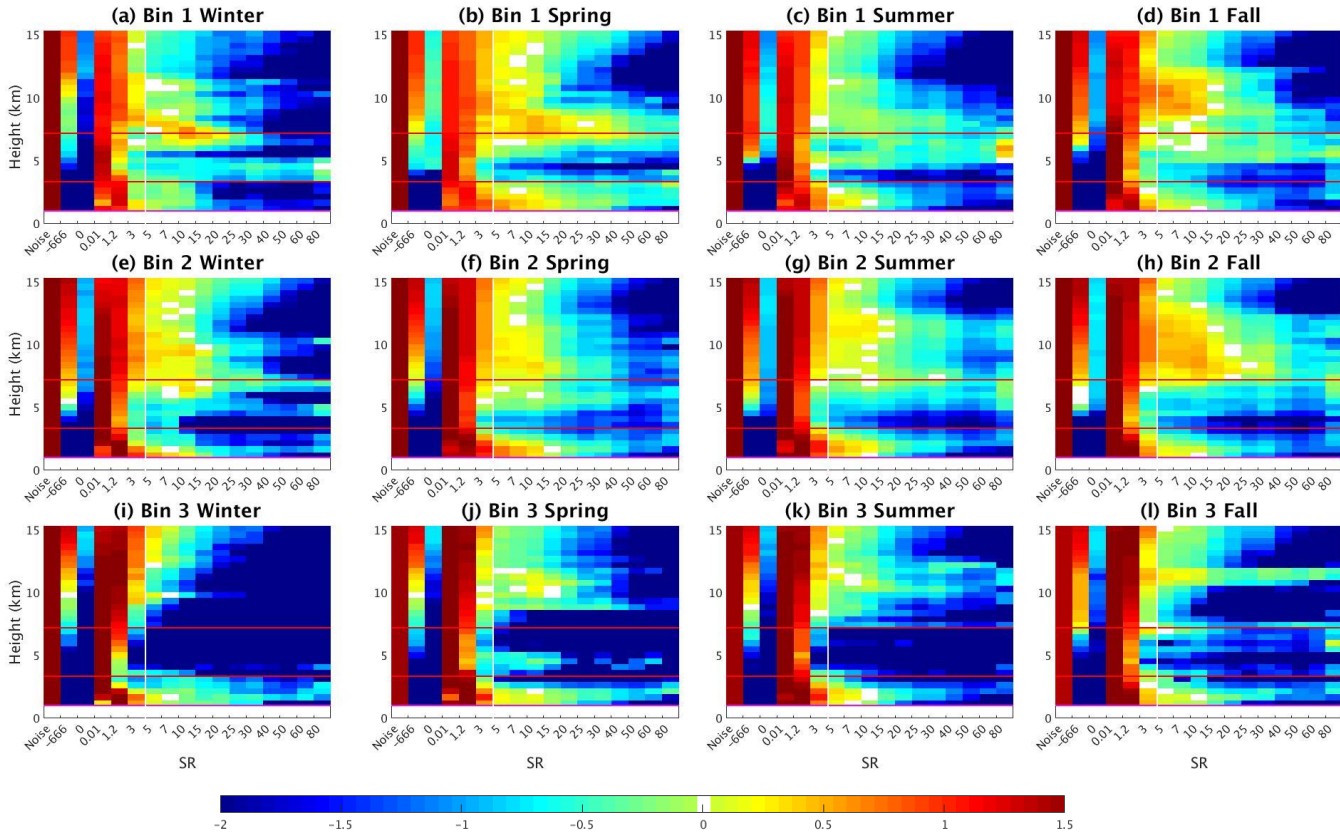

**Figure 9: Lidar scattering ratio (SR(z)) histogram obtained by cumulating all lidar observations during daytime for bin 1 (a)–(d), bin 2 (e)–(h) and bin 3 (i)–(l) for winter (first column), spring (second column), summer (third column) and fall (fourth column). The color bar is the logarithm of the percentage of occurrence (the sum of one level is equal to $log_{10}$ 100 %); lidar data showed in each subplot start above the instrument's recovery altitude (z = 1 km); the red horizontal lines represent the limits of low-, mid- and high-level clouds; and the white vertical line shows the threshold of clouds detection (SR(z) = 5).**





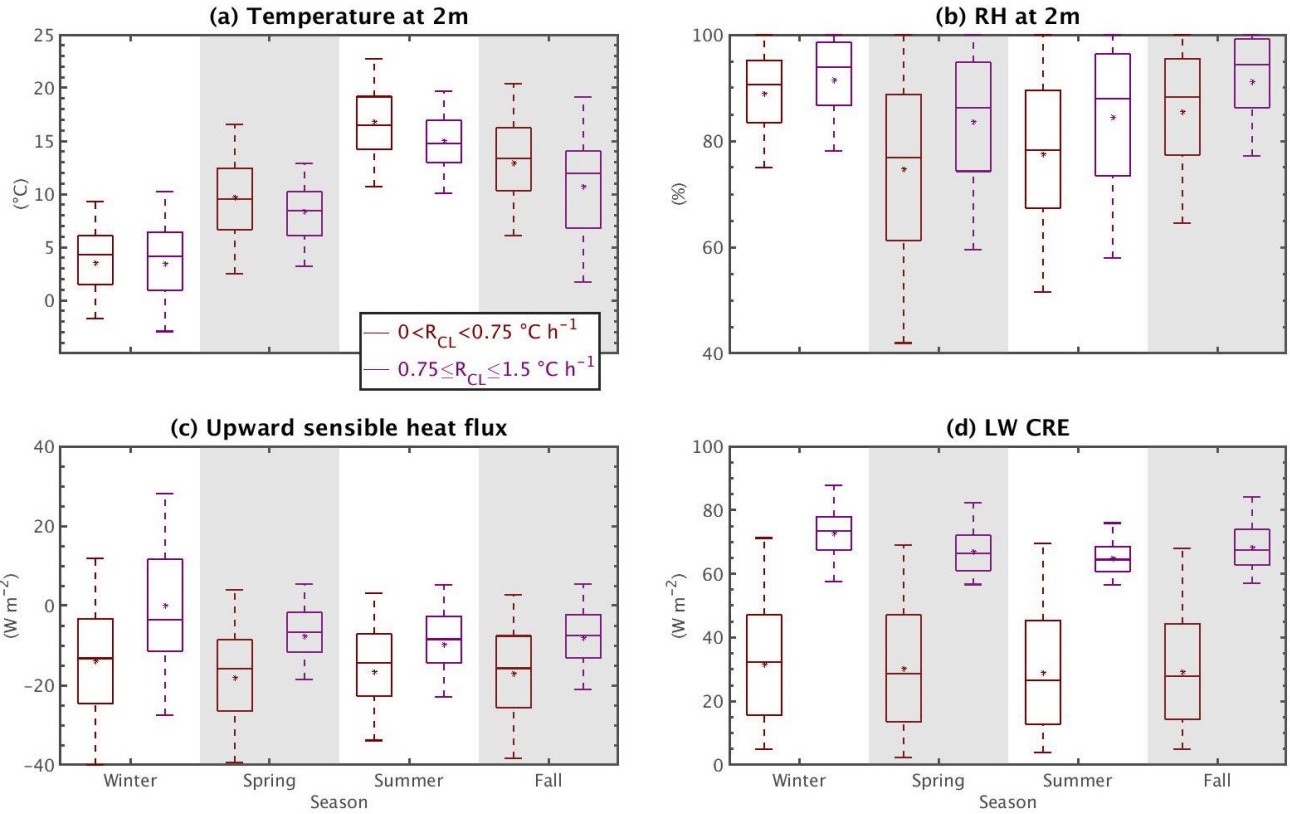


**Figure 10: Nighttime values for (a) temperature, (b) relative humidity, (c) upward sensible heat flux at 2 m, and (d) longwave cloud radiative effect. Colors and boxplots follow the same definition as in Fig. 7b.**






| Variable, unit | Notation | SIRTA-ReOBS available data from 2009 to 2014, in % |
|---|---|---|
| 2 m air temperature, K | $T_{2m}$ | 97 |
| Soil temperature below the ground, K | $T_s$ | 99 |
| Temperature at the mixing layer height*, K | $T_{MLD}$ | 71 |
| Surface downwelling LW radiation, W m$^{-2}$ | $F_{LW}^{\downarrow}$ | 100 |
| Surface downwelling SW radiation, W m$^{-2}$ | $F_{SW}^{\downarrow}$ | 100 |
| Surface upwelling LW radiation, W m$^{-2}$ | $F_{LW}^{\uparrow}$ | 91 |
| Surface upwelling SW radiation, W m$^{-2}$ | $F_{SW}^{\uparrow}$ | 94 |
| Surface downwelling LW radiation for clear sky, W m$^{-2}$ | $F_{LW,CS}^{\downarrow}$ | 97 |
| Surface downwelling SW radiation for clear sky, W m$^{-2}$ | $F_{SW,CS}^{\downarrow}$ | 97 |
| Surface upward sensible heat flux, W m$^{-2}$ | $F_{sens}$ | 73 |
| Surface upward latent heat flux, W m$^{-2}$ | $F_{lat}$ | 5 |
| Mixing layer depth, m | MLD | 71 |

Table 1: Variables available in the SIRTA-ReOBS dataset used as inputs in the temperature variation model. * $T_{MLD}$ is also estimated using the ERA5 Reanalysis dataset.

| Season | Correlation coefficient | Bias (°C h$^{-1}$) | Standard deviation (°C h$^{-1}$) |
|---|---|---|---|
| Winter | 0.67 (0.68) | -0.31 (-0.23) | 0.43 (0.40) |
| Spring | 0.80 (0.81) | -0.15 (-0.11) | 0.52 (0.45) |
| Summer | 0.82 (0.85) | -0.18 (-0.10) | 0.55 (0.46) |
| Fall | 0.79 (0.80) | -0.21 (-0.10) | 0.46 (0.41) |


Table 2: Statistics values between $\frac{\partial T_{2m}}{\partial t}_{obs}$ and $\frac{\partial T_{2m}}{\partial t}_{mod}$ for the four seasons of the year. The values in brackets correspond to the statistics within the 5th and 95th percentiles.




**Appendix: Detailed description of each term of the prognostic variations' temperature model at SIRTA observatory**

The prognostic model used to study the temperature change at the surface, considering all the components driving the surface energy balance, can be simply written as the sum of four processes:

$$\frac{\partial T_{2m}}{\partial t} = R + HG + HA + Adv \qquad (A1)$$

Where R is defined as the radiative forcing term, HG as ground heat exchange term, HA as atmospheric heat exchange term, and Adv as advection. Each of these terms represents a process involved in the variation of surface temperature $\frac{\partial T_{2m}}{\partial t}$, and can be calculated, using different measured and available variables as follows:

$$\frac{\partial T_{2m}}{\partial t} = \frac{\alpha+1}{\rho c_p MLH}\Delta F_{NET} + \frac{T_s - T_{2m}}{\tau_s} + \frac{T_{MLD} - T_{2m}}{\tau_a} - \left(u_{10}\frac{\partial T_{2m}}{\partial x} + v_{10}\frac{\partial T_{2m}}{\partial y}\right) \qquad (A2)$$

Where $T_{2m}$ is the surface temperature, $t$ is the time, $\alpha$ is a coefficient characterizing the form of the temperature profile in the
boundary layer, $\rho$ is the average air density of the boundary layer, $c_p$ is the specific heat of air, MLD is the height of the boundary layer, $\Delta F_{NET}$ is the net radiative flux at the surface, $T_s$ is the temperature in the ground at 20 cm depth, $T_{MLD}$ is the temperature at the top of the boundary layer, $u_{10}$ and $v_{10}$ are the zonal and meridional wind components, respectively, at 10m above the ground, and $\tau_s$ and $\tau_a$ are defined as relaxation timescales for heat exchange processes in the ground and the atmosphere, respectively.

Before explaining how each term is estimated, considerations are necessary about the stability of the atmosphere and related temperature profiles. Note that these assumptions will not affect the physical behavior of the method developed; they are made to have a more quantify treatment of the study.

Temperature near the surface and its behavior in lower layers are mostly driven by the quantity of net radiative flux arriving on it, which also depends on what moment of the day the temperature is observed; thus, it is important to differentiate day and
night. During the first case mentioned, radiative flux is significant as a result of solar incoming flux radiation (shortwave radiation), and the vertical temperature profile will depend on the amount of this radiative flux (among other components). In the absence of any incoming shortwave radiation, which corresponds to nighttime, clouds and other processes control temperature variations at the surface and its vertical profile.

Figure A1 confirms that the Planetary Boundary Layer (PBL) is on average unstable during daytime and stable during
nighttime in the area of study. This figure is built from twice-daily radiosoundings (at 11:00 and 23:00 local time) observations available in the SIRTA-ReOBS file from a METEO-France station located in Trappes (48.77° N, 2.01° E), 16 km away from the SIRTA observatory, to retrieve (every 15 m) the temperature and pressure up to 15 km above the ground. Figure A1a and b provide an overview of temperature profiles retrieved from these radiosoundings at 11:00 and 23:00 LT, respectively, for July of 2011 (each color represents a day of the month). Then, the monthly mean temperature profiles from 2003 to 2017, each
month represented as one color, are plotted in Fig. A1c and A1d. As expected, the temperature is decreasing from the surface





to the atmosphere in both daytimes' subfigures (Fig. A1a and A1c), and a temperature inversion occurs at lower layers in nighttime for the night cases (Fig. A1b and A1d) of the monthly mean.

This said, a daytime and a nighttime equation is used to parametrize the temperature profile in the surface layer (SFL).

For the daytime approach, a linear temperature profile is established as follows:

$$T(z) = T_{2m} + \beta * z \tag{A3}$$

Where $\beta$ is the temperature gradient, negative during day.

For the nighttime stable case, the temperature profile can be defined by a polynomial approach (Stull, 1988):

$$T(z) = T_{MLD} - \left(1 - \frac{z}{MLD}\right)^{\alpha} (T_{MLD} - T_{2m}) \tag{A4}$$

A shape parameter $\alpha = 1.5$ shows a quasi-linear behavior of the temperature profile with a weak positive gradient near the

surface, fitting well with the profiles in Fig. A1c and A1d (not shown).

The assumption of working with an ideal gas in an isobar environment, and applying the second law of thermodynamics, yield to an expression of the enthalpy as described below (Malardel, 2009; Wallace and Hobbs, 2006):

$$h = c_p T \tag{A5}$$

Knowing the vertical temperature profile, Eq. (A5) can be integrated all over the MLD to find the total enthalpy per square

meter at a given level (assuming that the density of the air is approximately constant at lower layers).

It is found that for a nighttime case, the total enthalpy can be expressed as:

$$h_{tot} = \rho c_p MLD \left[\frac{\alpha}{\alpha+1} T_{MLD} + \frac{1}{\alpha+1} T_{2m}\right] \tag{A6}$$

By deriving this equation by the surface temperature and total enthalpy, the ability to assess the change in surface temperature per unit change in enthalpy is estimated as:

$$\frac{\partial T_{2m}}{\partial h_{tot}} = \frac{\alpha+1}{\rho c_p MLD} \tag{A7}$$

For the daytime, proceeding as above, it is found the same equation for the nighttime with the only exception that $\alpha = 0$.

As explained before, distinguishing between night and daytime is very important for the study due to the differences in the radiative flux arriving at the surface, which condition the behavior of the PBL (i.e. MLD) and SEB terms.

**A1.1 Radiative term**

This term is calculated as:

$$R = \frac{\alpha+1}{\rho c_p MLD} \Delta F_{NET} \tag{A8}$$

The contribution of the radiative forcing on the temperature variations at SIRTA can be estimated by using the second law of thermodynamics, which states that the only energy a particle exchange with its surroundings is heat. That said, it is assumed that the only heat exchanged for a particle of air in the low atmosphere with its surrounding is the net radiative flux divergence,

yielding to:





$$\frac{\partial h_{tot}}{\partial t} = F^{\downarrow} - F^{\uparrow} \tag{A9}$$

Where $F^{\downarrow} - F^{\uparrow}$ is the only source of energy of the particle. By replacing $\partial h_{tot}$ in Eq. (A7) by (A9), it is found that:

$$\frac{\partial T_{2m}}{\partial t} = \frac{\alpha+1}{\rho c_p MLD}\left(F^{\downarrow} - F^{\uparrow}\right) \tag{A10}$$

$$\frac{\partial T_{2m}}{\partial t} = \frac{1}{\rho c_p MLD}\left(F^{\downarrow} - F^{\uparrow}\right) \tag{A11}$$

With Eq. (A10) and (A11) corresponding to night and daytime cases, respectively. These two equations described above will allow estimating the first term of the temperature variations model at SIRTA, the radiative term. MLD is a scale of height corresponding here to the height of the PBL retrieved from SIRTA-ReOBS. This value is set at this threshold because all the turbulent processes affecting the temperature variations are found within this layer. The PBL thickness varies depending, among other processes, on the amount of solar radiation, which enhances thermal and turbulent processes, making this

thickness ranges from tens of meters to 2 km or more. Thus, an average PBL thickness value (i.e. MLD) in an hourly and monthly scale is set (an assumption that doesn't affect the physical behavior of the model), as seen in Fig. A1c. Since during nighttime the boundary layer depth is difficult to estimate because of the weak turbulence due to the absence of solar radiation and its very complex dynamical system (Shi et al., 2005; Walters et al., 2007; McNider, 2011), a fixed value of 350 m is set (Fig. A1d). Figure A2b shows the behavior of the radiative term calculated by using Eq. (A12) and (A13), assuming $\rho = 1$ kg

m$^{-3}$ and $c_p = 1006$ J kg$^{-1}$ K$^{-1}$. A seasonal cycle is marked for this term, where during summers a peak of maximum contribution is found, whereas in winter its impact on temperature variations decreases significantly.

**A1.2 Atmospheric heat exchange term**

This term is estimated as:

$$HA = \frac{T_{MLD} - T_{2m}}{\tau_a} \tag{A12}$$

In this equation, $\tau_a$ can be calculated at equilibrium by setting the left side of the Eq. (A2) to zero and neglecting the ground heat exchange and advection terms (i.e. for the temperature not to vary, a balance between these two latter terms must be set), leading to:

$$\tau_a = \frac{\rho c_p MLD}{(\alpha+1)} * \frac{(T_{MLD} - T_{2m})}{\Delta F_{NET}} \tag{A13}$$

For the atmospheric heat exchange, $\Delta F_{NET} = F_{lat} + F_{sens}$. However, $F_{lat}$ has lots of missing values in SIRTA-ReOBS dataset

(material defection), and therefore does not allow performing a complete analysis for the five years of study. Hence, an hourly and monthly look-up table (LUT) is created to have an estimation of $\tau_a$. Moreover, $T_{MLD}$ is not directly available in the SIRTA-ReOBS dataset. To retrieve that temperature, the mixing layer depth is first retrieved from the SIRTA-ReOBS dataset, by looking for the nearest radiosounding in time (two per day) and founding the pressure corresponding to this altitude, then it is possible to have the temperature at that pressure levels by looking them in the ERA5 reanalysis dataset. Figure A3 shows an

average $\tau_a$ for all the months of the year. During night, the relaxation time scale shows a higher month-to-month variability



mainly due to the absence and gaps of the latent and sensible heat fluxes, which does not allow to reflect a marked tendency of $\tau_a$, whereas for daytime a more clearly behavior of $\tau_a$ is spotted for all the months thanks to an increase of the availability of the data for these hours (not shown). Sometimes during nighttime, this relaxation time exceeds 30 h, probably because the turbulence is so weak at that time that the heat does not reach completely the surface boundary layer (SBL) established. Stull

(1988) suggested that for the night cases, values of $\tau_a$ are on the order of 7 to 30 h, depending on the state of the SBL. Thus, values are limited to 30 h. For daytime, $\tau_a$ remains smaller because the turbulence is stronger thanks to convection occurring near the surface, leading to a faster communication of surface information across the lower layers.

The atmospheric heat exchange term hourly values are shown in Fig. A2d. Most of the values are negative, indicating that this term will modulate the positive contribution made by the radiative forcing term.

**A1.3 Ground heat exchange term**

The ground heat exchange term is defined as the energy lost by heat conduction through the lower boundary. This term can be then determined as follows:

$$HG = \frac{T_s - T_{2m}}{\tau_s} \tag{A14}$$

On average, the surface temperature during daytime (nighttime) is higher (lower) than that at low depth, as it's been shown

previously (Al-Hinti et al., 2017; Popiel and Wojtkowiak, 2013). Hence, the assumption that the vertical profile of temperature within the ground at low depths follows approximatively the same behavior as the atmospheric temperature profile can be established, and therefore the approach adopted for the atmospheric heat exchange term (c.f. A1.2) can be here implemented, to evaluate the relaxation time scale for the ground heat exchange at a depth of $H_s = 0.2$ m below the ground. It is considered the surface to be warmer than the temperature at 20 cm below the ground during the day thanks to the solar radiation arriving.

The opposite happens during night, when the ground losses important longwave radiation and it gets cold faster than $T_s$. Knowing the temperatures at the top of the ground ($T_{2m}$) and at 20 cm below it ($T_s$), the relaxation time scale results in:

$$\tau_s = \frac{\rho_s c_{p,s} H_s}{(\alpha + 1)} * \frac{(T_s - T_{2m})}{\Delta F_{NET}} \tag{A15}$$

The soil type at the SIRTA observatory is a mix of clay and limestone (obtainable from a regional predominant type of soil map in France; Wulf et al., 2015). The approximative values for ground density $\rho_s$ and heat capacity $c_{p,s}$ for this type of soil

are then:

$$\rho_s = 1300 \text{ kg m}^{-3}$$

$$c_{p,s} = 1140 \text{ J kg}^{-1} \text{ K}^{-1}$$

A mean value of $\tau_s = 20$ h for the day and night cases is found. Figure A2c presents the contribution of this term to the temperature variations, which is very minor but remains important to have a better agreement between the developed model

and the directly measured observations, by the fact that adding or removing some units could make a difference in estimating the correlation between the two datasets.



## A1.4 Advection term

$$Adv = \left( u_{10} \frac{\partial T_{2m}}{\partial x} + v_{10} \frac{\partial T_{2m}}{\partial y} \right) \tag{A16}$$

The advection term is estimated as presented in Eq. (A16) using ERA5 Reanalysis dataset of horizontal wind components at
10 m and temperature at 2 m above the ground. The purpose of using this dataset is to have another point in each horizontal axis to calculate the transport of the mass of air in zonal and meridional directions, between the SIRTA observatory and the immediately following grid box. For the period and time scale considered, the advection term plays in the majority a minor role, as shown in Fig. A2e, compared to the other terms.

## A1.5 Observed hourly temperature variations term

The left side of Eq. (A2) is defined as:

$$\frac{\partial T_{2m}}{\partial t} = \frac{T_h - T_{h-1}}{h} \tag{A17}$$

Where $T_h$ is the temperature at 2 m above the surface at hour $h$ and $T_{h-1}$ is the temperature at the previously considered hour. Figure A2a presents the evolution of this term, denoting, as expected, a seasonal cycle. Temperature variations have stronger negative than positive values, reaching few times a rate of -6 °C h$^{-1}$ or even -8 °C h$^{-1}$ for someday in summer 2010.

The sum of all terms on the right side of Eq. (A13) is hereafter called $\frac{\partial T_{2m}}{\partial t}_{mod}$.





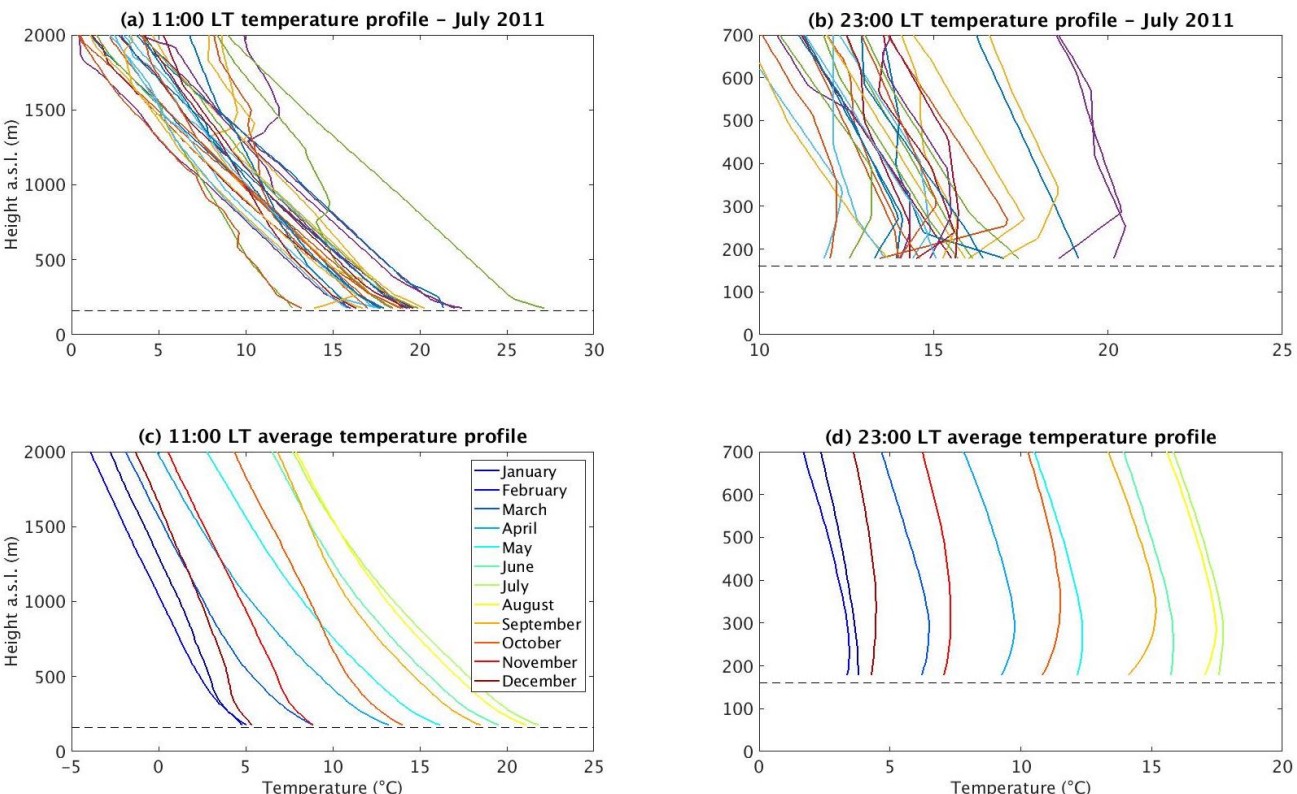

**Figure A1: Temperature profiles at 11:00 (a) and 23:00 (b) LT for July of 2011, and monthly averaged temperature profiles from 2003 to 2017 at 11:00 (c) and 23:00 (d) LT in Trappes. The temperatures at each altitude are retrieved from radiosoundings launched twice a day at the mentioned hours. The horizontal dashed line represents the altitude above sea level where the Trappes observatory is located, which is 160 m.**






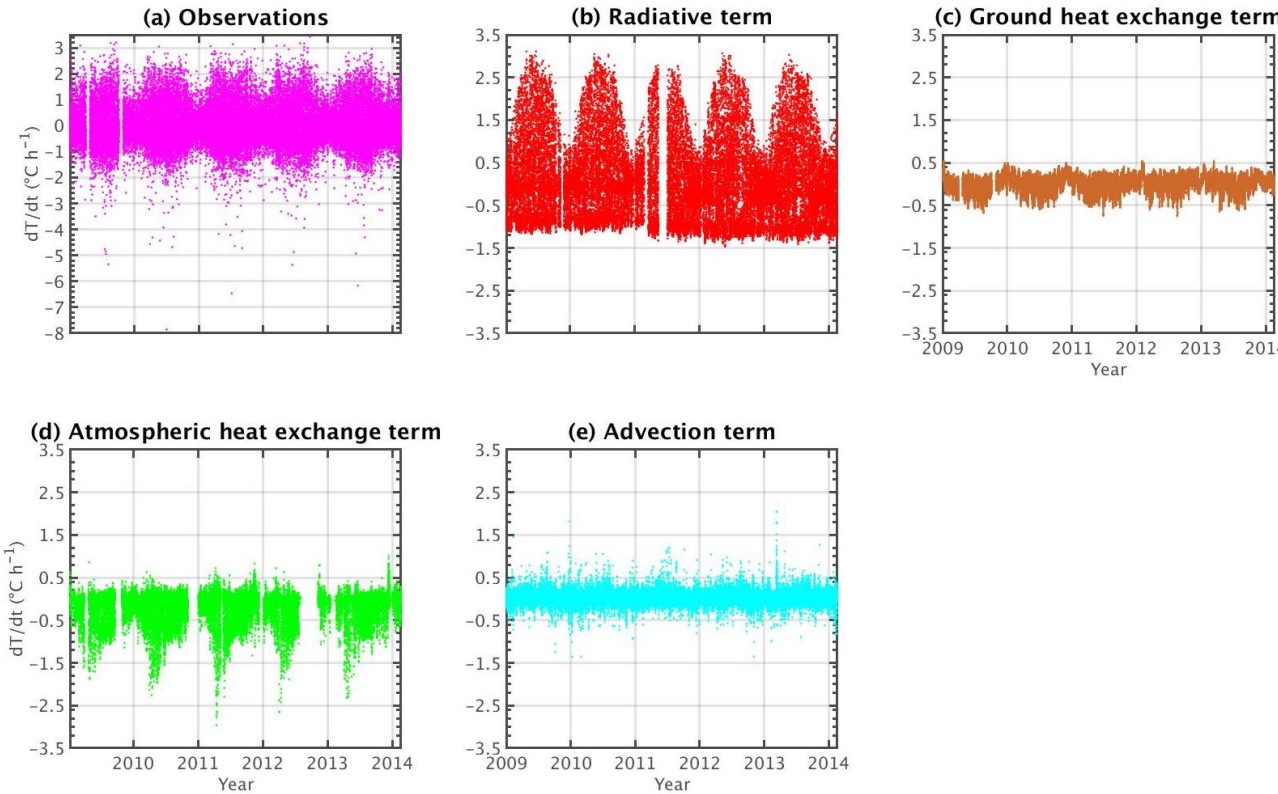

**Figure A2: Hourly evolution of observations (a), R term (b), the HG term (c), the HA term (d) and Adv term (e). Gaps in (d) are caused by the absence of MLD data at that moment.**





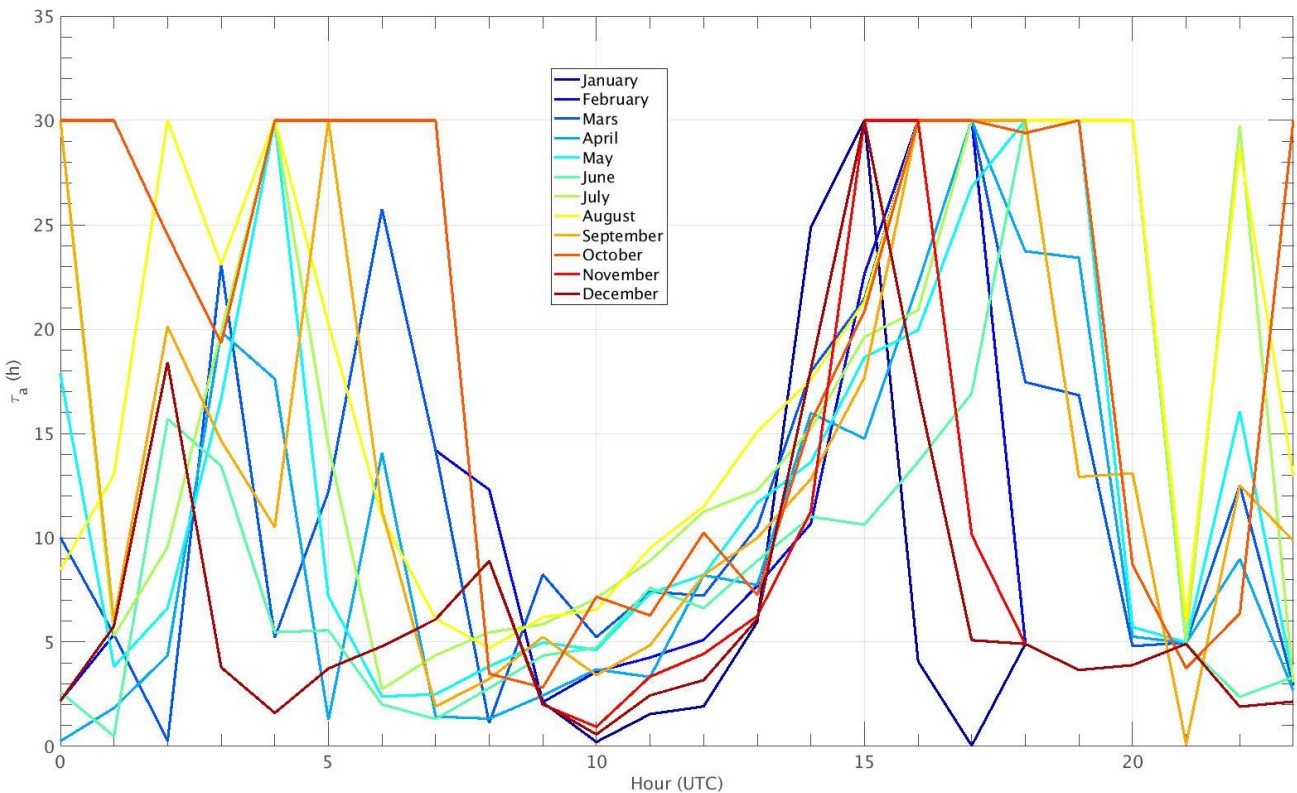


**Figure A3: Mean-hourly $\tau_a$ value for each month of the year.**