# Peer review of "Estimation of the terms acting on local surface one-hour temperature variations in Paris region: the specific contribution of clouds"

_Atmospheric Chemistry and Physics, 2021_

## Referee Comment (RC1)

**Review of the manuscript:**
**"Estimation of the terms acting on local surface one-hour temperature variations in Paris region: the specific contribution of clouds"**
**by Rojas et al.**

**General Comment**

In the paper "Estimation of the terms acting on local surface one-hour temperature variations in Paris region: the specific contribution of clouds" by Rojas et al., the authors develop an observation-based linear model to predict hourly temperature changes at SIRTA near Paris, France, to analyze the drivers of short-term temperature variability. The model uses surface energy budget terms to estimate temperature changes, and in an evaluation is in overall good agreement with observations. The authors have performed a detailed analysis of the contributions of individual surface energy budget terms (i.e. radiation, ground heat exchange, atmospheric heat exchange, and advection) for different times of day/year. A random forest model is applied to further study the influence of the individual terms. Finally, the influence of clouds is analyzed with more detail, using e.g. lidar observations.

Overall, this is a well written paper that could present a valuable contribution to the field, and the topic is of interest to the readership of ACP. The authors' conclusions are supported by the results displayed in the figures. In some parts of the manuscript, the descriptions are very detailed and rich (e.g. section 3 and the appendix), in other parts important information is missing (random forest analysis). I suggest this manuscript should be published in ACP after the comments have been adequately addressed.

**1 Specific comments**

l. 55–56 Results should not be mentioned in the introduction. Clouds are well known to modify near-surface air temperatures, which is justification enough to study them in more detail in this analysis.

l. 101 I suggest the authors use the higher-resolved ERA5 land (small differences in wind direction and the temperature fields may be relevant for temperature advection due to the vincinity of Paris). Also, it is not mentioned here that temperature data is also used from reanalysis (only in l. 915). Please add this information.

l. 140 I am guessing T2m is the near-surface air temperature and not the "surface temperature". Please correct this throughout the manuscript and appendix.

l. 270–289 There is nearly no information here on most aspects of the random forests model, which makes it impossible to reproduce the model, and hence the results, from the text.

- I am guessing that the model is trained to predict the observed temperature changes or is it the modeled temperature changes?
- What are the settings of the model, are the hyperparameters tuned, if so how?
- How is the data split up into training, testing and validation, what is the skill of the model in predicting temperature changes and is it overfitting the training data?
- The validation skill of the random forests model would be interesting - it should exceed the linear model if a) relations between predictors and the predictand are nonlinear or b) feature interaction effects help explain variability as hypothesized in l. 276. This should be tested and discussed.

l. 296–299 I don't quite understand this reasoning, as a) the authors use this approach to quantify the contributions of each term for all times of day in Fig. 2d), and b) the separate daytime/nighttime methods are used to calculate the individual terms used as predictors in the random forests model, right? Also, it is not clear to me how the authors derive the diurnal cycle of the feature importance during each season, this should be described in more detail in the manuscript.

l. 327–330 It would indeed be interesting to see if the temperature advection is wind-direction dependent. Is there a way to analyze the contribution of advection as a function of wind direction?

Fig. A1 The figures show typical daytime and nighttime temperature profiles for this region. What about sunrise and sunset, though? What uncertainties do these temperature profile regime transitions introduce at these times? This needs some discussion in the manuscript as diurnal cycles are investigated.

all Figs I think the quality of the figures should be improved by storing them as vector images instead of raster.

**2 Technical corrections**

l. 7 Maybe it would be good to clarify that you mean "Local short-term temperature variations".

l.12 Do you mean clear sky and cloudy sky?

l. 27 Please specify: variability of what?

l. 28–32 I think it would be good to be more precise on the temporal scales here.

l. 35 I suggest replacing "air advection" with "temperature advection"

l. 54–55 I suggest removing "whose maximal ... ones.", as random forests are used for many purposes and this statement is only true for some of them.

l. 62 and by time of day

l. 73 objectives cannot be answered

l. 74 "consists of describing" →"describes"

l. 86 "Southwest" →"southwest"

l. 145 x and y are not defined here (also missing in the appendix).

l. 307 I think you may want to change "modulate" to "dominate"

l. 340–341 This sentence needs to be corrected.

l. 354 I think it would be useful to show this histogram in the appendix.

l. 395–397 This seems to be a bit of an oversimplification of cirrus formation.

l. 489 This is speculative and a new aspect that should be discussed previously.

l. 494–497 This sentence is hard to understand.

l.530 The download links for the data should be provided in the acknowledgements.

l. 807 Please correct the grammar of this sentence.

l. 848 "exchange" →"exchanges"

---

## Author Comment (AC1)

**Response to Reviewer #1**

First of all, we would like to thank the first anonymous reviewer for the comments and rich suggestions, that help to improve our study.

**1. Specific comments**

**RC1-1: 55-56 Results should not be mentioned in the introduction. Clouds are well known to modify near-surface air temperatures, which is justification enough to study them in more detail in this analysis.**

Reviewer #2 also suggested to remove this sentence. This sentence was removed from the manuscript and replaced by:

*"Clouds are well known to modify directly near-surface temperatures and other near-surface variables in multiple time-scales"*

**RC1-2: 101 I suggest the authors use the higher-resolved ERA5 land (small differences in wind direction and the temperature fields may be relevant for temperature advection due to the vicinity of Paris). Also, it is not mentioned here that temperature data is also used from reanalysis (only in l. 915). Please add this information.**

The advection term is now estimated using the ERA5-Land Reanalysis, and thus all the Figures from 2 to 6 have been replaced with this new estimation. Now, we also mention in Line 644 that $T_{2m}$ is also retrieved from ERA5-Land in order to estimate the advection term. After performing a new analysis with this new advection term, results (and hence graphics) do not vary significantly but yet some analysis are changed with respect to Fig. 4e and Fig. 6, as well as Fig. A2e.
With respect to Figure 4e, the following has been added in Line 800-803:

*"The advection term does not present a strong monthly-hourly cycle compared to the other terms, although one can distinguish a mean negative action (still very low) to local temperature variations at all seasons with a mean minimum in July in the afternoon of -0.12 °C h$^{-1}$, as shown in Figure 4e."*

For Fig. 6, the following has been added in Line 894-898:

*"Regarding the Adv term, it shows an important weight in some hours in the late afternoon in winter, which makes it the term controlling on average hourly temperature variations at that time (then it is HA who becomes more important). In summer (Fig. 6c), it presents an important increase as the day goes on similar to HA after 10:00 UTC, but HA is even more important thanks to a development of turbulent heat fluxes at the surface in the late afternoon."*

In addition, all the graphics where this advection term was involved have been replaced in the manuscript (from Figure 2 to Figure 6).

**RC1-3: 140 I am guessing T2m is the near-surface air temperature and not the "surface temperature". Please correct this throughout the manuscript and appendix.**

Indeed, $T_{2m}$ is the near-surface air temperature, this is now corrected throughout all the manuscript and appendix.

**RC1-4: 270-289 There is nearly no information here on most aspects of the random forests model, which makes it impossible to reproduce the model, and hence the results, from the text**

- **RC1-4-1: I am guessing that the model is trained to predict the observed temperature changes or is it the modeled temperature changes?**

  In the original version of the paper, the random forest method is trained to predict the "modeled" temperature changes that correspond to the linear sum of the five terms, because the main objective of using random forest is to determine the importance of each term. This is now mentioned in the manuscript, see answer from **RC1-4-4** below.

  Nevertheless, in the new version of the paper, based on the feedback from the reviewers, we now also include the use of random forest method to predict the "observed" temperature variation (see following comments).

- **RC1-4-2: What are the settings of the model, are the hyperparameters tuned, if so how?**
- **RC1-4-3: How is the data split up into training, testing and validation, what is the skill of the model in predicting temperature changes and is it overfitting the training data?**
  The details of the settings of the random forest method along with the Figure A (Fig. B1 in the manuscript) are added in Appendix B, after the information on how the data is split up into training, testing, etc.:

*"**Appendix B: General information and basic settings on random forest method to study the weight of each term***

*Some hyperparameters are tuned in order to optimize the analysis:*

> *The random forest method is set to have 150 decision trees, because at that number the error converges to a small value, as seen in the Fig.B-1 (converging value of 0.25 during daytime, 0.12 during nighttime).*

[Figure]

**Figure A. Out-of-bag error over the number of grown regression trees for day (blue line) and night (orange line)**

> *For the split criteria, since our model is a simple one-degree regression, the method is set to use the mean square error (MSE) to do the split at each leaf.*

> ➢ *The number of random features to consider at each split and the number of bootstrapped dataset used to train each decision tree in the random forest method is approximatively to be 2/3 of the total of predictors and 2/3 of the total of sample, respectively (James et al., 2013)."*

*In Section 4, the random forest method is used to determine the importance of the predictors (terms) on the near-surface temperature variations. Detailed information on how this method works is given here.*

*The training algorithm for random forests applies the general technique of bagging, where the key to bagging is that trees are repeatedly fit to bootstrapped subsets of the observations. The bootstrapped term refers here to the fact of choosing randomly data that can be chosen several times to build decision trees (no selection restrictions). A training dataset is chosen randomly with replacement (bootstrapping) from the original dataset to create a decision tree. In regression techniques, the training dataset correspond to ~2/3 of the total of the sample. The ~1/3 remaining data not used to train that decision tree is used later as testing data but also to determine the importance of a specific term (James et al., 2013). This procedure is repeated for all the decision trees used in the random forest. Finally, it is not necessary to have a validation dataset in this study because the main interest of using this machine learning is to determine the importance of the terms on the model developed, as it is known that random forest protect against overfitting by constructing training samples through bootstrapping".*

However, following the reviewer requirements, the random forest method is now also used to predict new observed temperature variations. Further details are discussed in the answer from **RC1-4-4** review below.

- **RC1-4-4:** **The validation skill of the random forests model would be interesting - it should exceed the linear model if a) relations between predictors and the predictand are nonlinear or b) feature interaction effects help explain variability as hypothesized in l. 276. This should be tested and discussed.**

As stated in Line 841-844, a previous work (not shown) was carried out to analyze if there exists a linear relationship between each single term and the observed hourly temperature variations: this would allow to determine the importance of each term by looking at the slope of the linear regression between the single term and $\frac{\partial T_{2m}}{\partial t}_{obs}$ (as in Miller et al., (2017) at Summit, Greenland). In our case, no linear relationship was found between neither of the terms and $\frac{\partial T_{2m}}{\partial t}_{obs}$ hence the random forest method is suitable.

To answer to **RC1-4-1** and **RC1-4-3**, the following paragraph is added to the manuscript in Line 828-835:

*"One of the most impressive features of RF is here used, which consists on the ability to provide a fully nonparametric estimation of the importance of each term (or predictor) on the model. One of the main advantages of this method is that it allows covering not only the impact of each term individually in the model but also the multivariate interactions with other predictors. Here, the model (i.e. $\frac{\partial T_{2m}}{\partial t}_{mod}$ ) has been already developed and defined as the sum of five terms. Therefore, to determine the importance of each term, the input data (the five terms) in the RF method are trained to predict the modeled temperature changes ($\frac{\partial T_{2m}}{\partial t}_{mod}$), and so here the output of the RF method is still $\frac{\partial T_{2m}}{\partial t}_{mod}$ . To know more on how the hyperparameters are*

*tuned, how is the data split up into training and testing, and further information on the RF method, please refer to Appendix B."*

For the validation skill of the random forest method, the predicted values of $\frac{\partial T_{2m}}{\partial t}_{obs}$ after the data have been trained to predict the observed temperature variations, are now presented in Fig. B (Fig. 7 in the manuscript), and its analysis is added in a new sub-section, **Section 4.3 Validation of the random forest method** (Line 913-925), and discussed as follows:

*"However, this machine learning method is generally used in other studies to train and to have better estimations of a particular model. In order to validate the random forest method skill on predicting new observed temperature variations, the method is used to predict $\frac{\partial T_{2m}}{\partial t}_{obs}$ (rather than $\frac{\partial T_{2m}}{\partial t}_{mod}$ as it is done to estimate the weight of each term in Section 4.1). The output for this case is called $\frac{\partial T_{2m}}{\partial t}_{obs,RF}$. A comparison between $\frac{\partial T_{2m}}{\partial t}_{mod}$ (i.e. the linear sum of the five terms) and the new $\frac{\partial T_{2m}}{\partial t}_{obs,RF}$ is done, the results of this validation are shown in Fig. 7. Indeed, the scatterplot before performing the random forest method ($\frac{\partial T_{2m}}{\partial t}_{mod}$) shows the distribution of values between the observations and the model (i.e. $\frac{\partial T_{2m}}{\partial t}_{obs}$ vs $\frac{\partial T_{2m}}{\partial t}_{mod}$, blue points) as found in Fig. 2c. Then, when the random forest method is performed and the data are trained based on $\frac{\partial T_{2m}}{\partial t}_{obs}$ (instead of $\frac{\partial T_{2m}}{\partial t}_{mod}$), better predictions are obtained between $\frac{\partial T_{2m}}{\partial t}_{obs}$ and $\frac{\partial T_{2m}}{\partial t}_{obs,RF}$ (orange points) and the correlation coefficient has a higher value (0.94). In such case, the RF method gives better estimations of temperature variations but the retrieve of the function used to have these results is not available. Nevertheless, this result validates the fact of considering temperature variations as the sum of the five terms to estimate their importance using the RF method (when it is used to predict the modeled temperature variations)."*

[Figure]

**Figure B. Scatter plot of $\frac{\partial T_{2m}}{\partial t}_{obs}$ as a function of the developed model $\frac{\partial T_{2m}}{\partial t}_{mod}$ before applying the random forest method (blue circles) and $\frac{\partial T_{2m}}{\partial t}_{obs}$ as a function of the model trained after the RF method is applied (orange circles).**

**RC1-5:** 296-299 I don't quite understand this reasoning, as a) the authors use this approach to quantify the contributions of each term for all times of day in Fig. 2d), and b) the separate daytime/nighttime methods are used to calculate the individual terms used as predictors in the random forests model, right? Also, it is not clear to me how the authors derive the diurnal cycle of the feature importance during each season, this should be described in more detail in the manuscript

At first (Fig. 5), the random forest method is performed for all the daytime (resp. nighttime) hours together (one single calculation). Then (Fig. 6), the same analysis is performed but independently at each hour of the day and for each season. Thus, this sentence is replaced in Line 862-864 by:

"*The importance estimation previously calculated for both day and nighttime periods considers all the processes occurring during each case and thus gives a global importance estimation. In order to separate the influence of each term on hourly temperature variations, an importance estimation value is performed for each hour independently*"

To explain how the diurnal cycle of the feature importance is estimated during each season, the following statement has been added in Line 868-869:

"*Figure 6 presents the results of this method for each season. This diurnal cycle is estimated by applying the random forest to each hour separately.*".

**RC1-6:** 327-330 It would indeed be interesting to see if the temperature advection is wind-direction dependent. Is there a way to analyze the contribution of advection as a function of wind direction?

A wind distribution is estimated to assess how the wind direction affects the contribution of advection on hourly temperature variations. An advection rose distribution based on wind direction and advection contribution is plotted and showed in Fig. C (Not added to the manuscript).
This figure shows us that the negative contribution of the Adv term is mostly coming from N-E and S-E wind regimes, whereas the positive contributions are from S-W regime. Few percentages of winds are coming from N-W. Indeed, this term contributes strongly to cool the surface for a S-E regime. Figure C also illustrates the two majority winds regimes coming to the Paris region: the well-known Siberian High which brings on average cold winds and is in agreement with Fig. C, and the Westerly winds coming from the Atlantic Ocean with warmer and humid near-surface air.

Furthermore, whatever the wind direction, the advection term remains of the same order of magnitude in importance as HA and HG and much less important than $R_{CS}$ and $R_{CL}$ at this time scale, as shown in Fig. D (not added to the manuscript) when the random forest method is applied separately for each wind regime (N-E, S-E, S-W and N-W). The following statement has been added in Line 902-904 to summarize these results:

"*Indeed, in this area the two predominant winds come from S-E regime (the Siberian High) bringing mostly cold air temperatures, and S-W regime (air masses coming from the Atlantic Ocean) with warmer and more humid air (not shown).*"

[Figure]

**Figure C. Advection distribution at SIRTA from 2009 to February 2014.**

[Figure]

**Figure D. Predictor importance estimates obtained by the random forest method as a function of wind direction for the five terms of the model developed from January 2009 to February 2014.**

**RC1-7: Fig. A1. The figures show typical daytime and nighttime temperature profiles for this region. What about sunrise and sunset, though? What uncertainties do these temperature profile regime transitions introduce at these times? This needs some discussion in the manuscript as diurnal cycles are investigated**

The reviewer asks for temperature profiles at sunrise and sunset, which are not available at SIRTA or Trappes, since radiosoundings are launched twice a day at 11:00 and 23:00 LT. It is true indeed that it would be interesting to study these transition zones. To analyze this, contours indicating sunrise and sunset hours were added in all the subfigures in Figure E (Figure 4 in the manuscript). Our model seems to reproduce on average quite well $\frac{\partial T_{2m}}{\partial t}$ both at sunrise and sunset since the residual is low at these times (Fig. 4f). A new paragraph is added in Section 3.3, which is consecrated to evaluate these transitions periods (Line 815-818), which was also suggested by the Reviewer #2:

*"Focusing on the transition periods (sunrise and sunset, black lines in Fig. 4), the residual presents low values at these times. Indeed, there is a slight underestimation of the model of about -0.13 °C h⁻¹ for some months (e.g. February) at sunrise hours, whereas a low overestimation with close-to-zero residual mean values are found for May and June. For the sunset, a similar behavior is found (with very similar values for the residual term). Therefore, a good agreement is found between the model and the observations for these specific hours."*

[Figure]

**Figure E:** Monthly-hourly mean values for (a) $R_{CS}$, (b) $R_{CL}$, (c) HG, (d) HA, (e) Adv and (f) the residual (i.e. difference between the model and the observations). Units on the color bars are all in $°C\ h^{-1}$, and their scale is different for each subfigure. The black contour line on each figure corresponds to sunrise (bottom line) and sunset (top line) approximative hours.

**RC1-8: I think the quality of the figures should be improved by storing them as vector images instead of raster**

When storing the figures as vector images the quality does not improve (e.g. with a .svg extension), and when we try to store them as .eps images, the last version of Microsoft Word 2019 does not support this extension anymore.

**2. Technical corrections**

**RC1-9:** **7 Maybe it would be good to clarify that you mean" Local short-term temperature variations".**

Sentence corrected and replaced by "*Local short-term temperature variations*".

**RC1-10:** **12 Do you mean clear sky and cloudy sky?**

Yes, now it is corrected in the abstract

**RC1-11:** 27 Please specify: variability of what?

This phrase was modified as:

*"...controls the air mass advection over western Europe and explains a large part of weather variability"*

**RC1-12:** 28-32 I think it would be good to be more precise on the temporal scales here.

In order to mention the temporal scales affecting in first order temperature and pressure conditions, the following new references have been added which state how temperatures and precipitation are affected by interannual atmospheric circulations:

Efthymiadis, D., Goodess, C. M., and Jones, P. D.: Trends in Mediterranean gridded temperature extremes and large-scale circulation influences, 11, 2199–2214, https://doi.org/10.5194/nhess-11-2199-2011, 2011.

Xoplaki, E., González-Rouco, J., Gyalistras, D., Luterbacher, J., Rickli, R., and Wanner, H.: Interannual summer air temperature variability over Greece and its connection to the large-scale atmospheric circulation and Mediterranean SSTs 1950–1999, Climate Dynamics, 20, 537–554, https://doi.org/10.1007/s00382-002-0291-3, 2003.

Xoplaki, E., González-Rouco, J. F., Luterbacher, J., and Wanner, H.: Wet season Mediterranean precipitation variability: influence of large-scale dynamics and trends, Climate Dynamics, 23, 63–78, https://doi.org/10.1007/s00382-004-0422-0, 2004.

Bartolini, E., Claps, P., and D'Odorico, P.: Interannual variability of winter precipitation in the European Alps: relations with the North Atlantic Oscillation., 13, 17–25, https://doi.org/10.5194/hess-13-17-2009, 2009.

Furthermore, Line 553-557 have been modified and the following statement is added:

*"Temperature and pressure conditions are then modulated by the complex terrain (Mediterranean sea, topography, surface heterogeneities): extreme events and temperature anomalies are generally not exclusively explained by the presence of these large-scale air mass circulations (Vautard and Yiou, 2009). Indeed, synoptic and meso-scale atmospheric processes have been previously studied to explain interannual temperature changes in some parts of Europe (Efthymiadis et al., 2011; Xoplaki et al., 2003), or even precipitation occurrence (Xoplaki et al., 2004; Bartolini et al., 2009)."*

**RC1-13:** 35 I suggest replacing "air advection" with "temperature advection"

Air advection is replaced by temperature advection in that part of the manuscript

**RC1-14: 54-55 I suggest removing "whose maximal … ones.", as random forests are used for many purposes and this statement is only true for some of them**

This adjective maximal is removed and replaced by *"whose one of its attributes is …"*

**RC1-15:** 62 and by time of day

The phrase is corrected and now it is:

*"But their damping effects vary on the season and by time of day"*

**RC1-16:** 73 objectives cannot be answered

The verb "to answer" has been replaced by "to achieve"

**RC1-17:** 74 "consists of describing"→"describes

This is corrected on the manuscript

**RC1-18:** 86 "Southwest"→"southwest"

Spell is corrected

**RC1-19:** 145 x and y are not defined here (also missing in the appendix).

X and y now are defined in that section and in the appendix, as follows:

*"x is the zonal wind component towards the east and y is the meridional wind component towards the nord"*

**RC1-20:** 307 I think you may want to change "modulate" to "dominate"

The word "modulate" is changed by "dominate" as the reviewer suggested

**RC1-21:** 340–341 This sentence needs to be corrected

This sentence is now corrected to:

*"Knowing how and in what measure each term contributes to temperature variations, a deeper analysis is performed in this section in order to better understand the role of clouds"*

**RC1-22:** 354 I think it would be useful to show this histogram in the appendix

The histograms for both day and nighttime cases used to create the bins to study the role of clouds are presented in Fig. F and in the Appendix C, Fig. C1 in the manuscript. A new sentence is added in Line 949 specifying this, as follows:

*"This histogram, along with the one used in Section 5.2, are presented in Appendix C."*

[Figure]

**Figure F. (a) Daytime histogram of $\frac{R_{CL}}{R_{CS}}$ and (b) nighttime histogram of $R_{CL}$. The red line in both figures represents the PDF and the rectangular semi-transparent brown boxes the different bins created to analyze clouds influence. These histograms are built by considering only cloudy hours. Negative and close-to-zero values are removed for daytime hours (see text for further information).**

**RC1-23:** 395–397 This seems to be a bit of an oversimplification of cirrus formation.

Indeed, here we try to be very synthetized on how cirrus clouds are formed (the scope of our study not being the detail study of clouds microphysics). The reason stated in the manuscript correspond to one possible explanation to the presence of these clouds for the two seasons mentioned, but it does not remain the ultimate reason why we see these clouds. This sentence has been modified as follows:

*"Indeed, one reason explaining the presence of these high-level clouds at these two transition seasons could be the meet of a warm air with a cold air mass (which occur more often at spring and fall), where the lighter warm air rises up to several km from the ground and could form some cirrus clouds"*

**RC1-24:** 489 This is speculative and a new aspect that should be discussed previously

The possible presence of these clouds is now discussed in Section 5.1.1. when a great SWCRE is found for the bin having a strong cooling effect of clouds (bin 1), and therefore added in Line 980-982:

*"(…). These strong and negative SWCRE could be associate with a presence of nimbostratus clouds, due to the high SR detected for lidar, clouds which are more likely to form in summer because of the strong convective systems developed during that time due to higher surface temperatures."*

**RC1-25:** 494–497 This sentence is hard to understand

The mentioned sentence has been better written as follows:

*"In addition, situations with weak cloud effect (either negative or positive) coincide with an important amount of high-level thick clouds for all the seasons (except winter) whose LWCRE is high, but SW clear*

*sky radiation controls temperature variations (Fig. 8a, bins 2 and 3). These high-level clouds are more present in weak cloud cooling and warming effects (bins 2 and 3) than the times of strong cooling effect (bin 1)"*

**RC1-26:** 530 The download links for the data should be provided in the acknowledgements

The link to download the SIRTA-ReOBS dataset is now provided in the acknowledgements

**RC1-27:** 807 Please correct the grammar of this sentence.

Grammar is corrected and now this sentence is:

*"Note that these assumptions will not affect the physical behavior of the developed method; they are made in order to have a more quantitative treatment of the study"*

**RC1-28:** 848 "exchange"→"exchanges"

The grammar is now corrected

---

## Author Comment (AC2)

**Response to Reviewer #2**

We would like to thank the anonymous reviewer for these suggestion and relevant comments.

**RC2-1: Abstract: First two lines fit well for an introduction. What is the overall major scientific problem regarding the goal of this manuscript should be first spelled out in the first part of the abstract, instead of generic information on surface temperature variability?**

The first sentence presented in the abstract has been changed in order to present the major scientific problem and scope of the current study, as follows:

"*Local short-term temperature variations at the surface are mainly dominated by small-scale processes coupled through the surface energy balance terms, which are well known but whose specific contribution and importance on the hourly scale still need to be further analyzed*"

**RC2-2:** Intro: Authors should clearly mention the need for such a model which is based on a lot of observations. How can this model help develop and improve surface layer parameterization scheme. Etc

Now this is mentioned in the introduction in Line 579-588 as follows:

"*The use of the model developed in the current study considers all the variables acting within the ABL and controlling surface temperature variations, all of them estimated almost exclusively from surface-based observations. Thus, it allows to study separately the influence of each SEB term in a local scale. This indeed allows to have a realistic and reliable estimation of the contribution of each term (radiative fluxes, turbulent heat fluxes, etc.) on hourly temperature variations, and it would be possible to have that at different sites since each term will present a different behavior and importance. These estimations could help improving the parametrizations already existing of the SEB terms and better understanding their spatial evolution as a function of local conditions. Furthermore, a comparison between multi-model regional climate simulations and these estimations can be performed to evaluate if the simulations are able to well reproduce these behaviors, in particular in a warming climate where these processes are expected to change.*"

**RC2-3:** Line 28: Azores?

We suppose the reviewer meant Line 27 from the original manuscript, and maybe there is an error of encoding in the PDF from the reviewers PDF viewer since it is written "Açores"

**RC2-4:** Line 55: This is result and too early to spell out here. Please remove.

It is modified since the reviewer #1 also suggested to remove the results in the introduction, in Line 592-593:

"*Clouds are well known to modify directly near-surface temperatures and other near-surface variables in multiple time-scales (Parding et al., 2014; Broeke et al., 2006; Kauppinen et al., 2014).*"

**RC2-5:** Line 65: Set an example for each with references for "climate variability and extreme local events". I suggest 2006 drought in EU

Three references are mentioned now that show extreme local events influenced by the presence of clouds: Chiriaco et al., (2014); Rebetez et al., (2009); Bennartz et al., (2013)

**RC2-6:** Line 72: Repeated. Delete please.

It is deleted and then slightly modified

**RC2-7:** Line 105: Which lidar and what is the temporal and vertical resolution of lidar instrument here. Some details could be found in Koffi et al. (Evaluation of the boundary layer dynamics of the TM5 model over Europe) on different EU sites on this.

It is now specified the type of lidar used as well as its vertical resolution, and it is now mentioned in Line 649-650 that further details can be found in Chiriaco et al., (2018) in Table 1 and Section 3.5. This is mentioned as follows:

*"(…) retrieved from a LNA lidar (532 and 1064 nm) whose vertical resolution is 15 m (for further details, see Chiriaco et al., 2018), (…)"*

**RC2-8:** Line 118: Unless it has been established before, this is too early in a manuscript. Please remove.

Indeed, it is too early to mention it, so this sentence has been removed.

**RC2-9:** Line 171: Please use the term "combined"

The sentence is changed as follows:

*"… is retrieved using SIRTA-ReOBS combined with ERA5 dataset"*

**RC2-10:** Line 182: "mixing with an atmosphere of higher levels….". If so, then how does it represent a high positive correlation coefficient found in other literature where the authors have performed regression analyses of MLD and surface temperature. See Seidel et al.2010, 2012 (Climatology of the planetary boundary layer over the continental United States and Europe). This has an important implication. Please clarify. I think above statement need to justified and corrected.

Seidel et al. (2012) showed a strong correlation between MLD and surface temperature (especially in warm seasons), but this correlation is found for radiosondes launched at 12:00 UTC. Moreover, both the temporal and spatial scales are different from those of ours. Seidel et al. (2012) looked at a continental scale with measurements retrieved from several observatories in North America and Europe, whereas our study focuses on a local scale. As for the temporal scale, they performed an annual cycle of MLD and $T_{2m}$ and then found the strong correlation between these two variables. It is also expected that this high correlation is found for the rest of the day since the surface temperature keeps increasing as MLD does it as well. However, here we refer to the contribution of the HA to hourly surface temperature variations, not to surface temperature. The HA term presents a negative contribution to hourly surface temperature variations in the afternoon as shown in Figure A (figure not added to the manuscript) for all seasons, therefore we mention that the mixing of higher atmosphere levels indeed contributes to cool the surface. We change this sentence in Line 727-728 to clarify that we do refer to surface temperature variations, as follows:

*"(..) meaning that the mixing with an atmosphere of higher levels contributes to decrease surface temperature variations, even if surface temperatures continue increasing along the day."*

[Figure]

**Figure A. Diurnal cycle of the five terms of our model, and the observed temperature variations, split into seasons.**

**RC2-11:** Line 186: Please quantify (remains low).

This sentence is now corrected in Line 731-732:

*"…: differences occur for cases where the temperature decreases during the hour, but this difference corresponds to some cases where the model presents more negative values than the observations, around -1 °C h⁻¹"*

**RC2-12:** Figs. 2a and 2b: x-axes scale limits need to be symmetric; otherwise, one cannot justify the statements made in this regard.

Both x-axes on Figs. 2a and 2b have been modified and now they are symmetric

**RC2-13:** Line 189 and associated figure: Since observation is the reference here for the analytical model, please exchange the x and y axes of Fig. 2c.

Indeed, the linear regression that best fits the model for the observation is calculated and presented in Fig. 2c: $y$ corresponds $\frac{\partial T_{2m}}{\partial t}_{obs}$ and $x$ is $\frac{\partial T_{2m}}{\partial t}_{obs}$, so the best linear fit found is $\frac{\partial T_{2m}}{\partial t}_{obs} = p * \frac{\partial T_{2m}}{\partial t}_{mod} + b$, where $p$ is the coefficient and b the intersection with the y-axis. We found therefore convenient to represent in the y-axis the observations and in the x-axis the model

**RC2-14:** Fig. 4: Units are missing on the color bar scale limits. Please use symmetric color bar scalelimits as well like in Fig. 4f

Units of the color bar are specified on figure caption. Using symmetric color bar scales limits do not allow to have a clear view of the contribution of each term, the idea is also to set red colors as a positive contribution and blue as negative contributions. We use the upper and bottom scale limits as the maximum and minimum value contribution of each term for a clearer interpretation.

**RC2-15:** Section 4.2: It will be important so that the authors should focus on the analyses of temperature variability during morning and evening transition periods which are the two most complicated phases of the diurnal cycle of temperature over land and this is also important for trace gas variability as well since the ABL interacts with upper layers in phases (e.g., Lee et al. Meteorological controls on the diurnal variability of carbon monoxide mixing ratio at a mountaintop monitoring site in the Appalachian Mountains).For the above, I suggest rather than each hour temporal variability, author could build a key temperature growth rate (between sunrise and 14 UTC) and compare that single parameter in different seasons and years (model vs obs).

What the reviewer asked us to do is not possible to perform because we don't have the surface temperature estimated by the model. Nevertheless, to analyze if the model well reproduces the observed temperature variations,
contours indicating sunrise and sunset hours were added in all the subfigures in Figure B (Figure 4 in the manuscript). Our model seems to reproduce on average quite well $\frac{\partial T_{2m}}{\partial t}$ both at sunrise and sunset since the residual is low at these times (Fig. Bf ). This is also corroborated again in Fig. A that show us that the residual is on average weak (gray dashed line) at these transition hours (marked as the vertical black dashed lines) for all the four seasons.  A new paragraph is added in Section 3.3, which is consecrated to evaluate these transitions periods (Line 815-818):

"*Focusing on the transition periods (sunrise and sunset, black lines in Fig. 4), the residual presents low values at these times. Indeed, there is a slight underestimation of the model of about -0.13 °C h$^{-1}$ for some months (e.g. February) at sunrise hours, whereas a low overestimation with close-to-zero residual mean values are found for May and June. For the sunset, a similar behavior is found (with very similar values for the residual term). Therefore, a good agreement is found between the model and the observations for these specific hours.*"

[Figure]

**Figure B: Monthly-hourly mean values for (a) $R_{CS}$, (b) $R_{CL}$, (c) HG, (d) HA, (e) Adv and (f) the residual (i.e. difference between the model and the observations). Units on the color bars are all in °C h$^{-1}$, and their scale is different for each subfigure. The black contour line on each figure corresponds to sunrise (bottom line) and sunset (top line) approximative hours.**